# AGD: an Auto-switchable Optimizer using Stepwise Gradient Difference for Preconditioning Matrix

**Yun Yue** [*]
Ant Group
Hangzhou, Zhejiang, China
yueyun.yy@antgroup.com

**Zhiling Ye** [*]
Ant Group
Hangzhou, Zhejiang, China
yezhiling.yzl@antgroup.com

**Jiadi Jiang** [*]
Ant Group
Hangzhou, Zhejiang, China
jiadi.jjd@antgroup.com

**Yongchao Liu**
Ant Group
Hangzhou, Zhejiang, China
yongchao.ly@antgroup.com

**Ke Zhang**
Ant Group
Beijing, China
yingzi.zk@antgroup.com

## Abstract

Adaptive optimizers, such as Adam, have achieved remarkable success in deep learning. A key component of these optimizers is the so-called preconditioning matrix, providing enhanced gradient information and regulating the step size of each gradient direction. In this paper, we propose a novel approach to designing the preconditioning matrix by utilizing the gradient difference between two successive steps as the diagonal elements. These diagonal elements are closely related to the Hessian and can be perceived as an approximation of the inner product between the Hessian row vectors and difference of the adjacent parameter vectors. Additionally, we introduce an auto-switching function that enables the preconditioning matrix to switch dynamically between Stochastic Gradient Descent (SGD) and the adaptive optimizer. Based on these two techniques, we develop a new optimizer named AGD that enhances the generalization performance. We evaluate AGD on public datasets of Natural Language Processing (NLP), Computer Vision (CV), and Recommendation Systems (RecSys). Our experimental results demonstrate that AGD outperforms the state-of-the-art (SOTA) optimizers, achieving highly competitive or significantly better predictive performance. Furthermore, we analyze how AGD is able to switch automatically between SGD and the adaptive optimizer and its actual effects on various scenarios. The code is available at this link[2].

## 1 Introduction

Consider the following empirical risk minimization problems:

$$\min_{\boldsymbol{w} \in \mathbb{R}^n} f(\boldsymbol{w}) := \frac{1}{M} \sum_{k=1}^{M} \ell(\boldsymbol{w}; \boldsymbol{x}_k), \tag{1}$$

where $\boldsymbol{w} \in \mathbb{R}^n$ is the parameter vector to optimize, $\{\boldsymbol{x}_1, \ldots, \boldsymbol{x}_M\}$ is the training set, and $\ell(\boldsymbol{w}; \boldsymbol{x})$ is the loss function measuring the predictive performance of the parameter $\boldsymbol{w}$ on the example $\boldsymbol{x}$. Since it is expensive to calculate the full batch gradient in each optimization iteration when $M$ is large, the

---

[*]Co-first authors with equal contributions.

[2]https://github.com/intelligent-machine-learning/dlrover/tree/master/atorch/atorch/optimizers

37th Conference on Neural Information Processing Systems (NeurIPS 2023).

standard approach is to adopt a mini-batched stochastic gradient, i.e.,

$$\boldsymbol{g}(\boldsymbol{w}) = \frac{1}{|\mathcal{B}|} \sum_{k \in \mathcal{B}} \nabla \ell(\boldsymbol{w}; \boldsymbol{x}_k),$$

where $\mathcal{B} \subset \{1, \ldots, M\}$ is the sample set of size $|\mathcal{B}| \ll M$. Obviously, we have $\mathbf{E}_{p(\boldsymbol{x})}[\boldsymbol{g}(\boldsymbol{w})] = \nabla f(\boldsymbol{w})$ where $p(\boldsymbol{x})$ is the distribution of the training data. Equation (1) is usually solved iteratively. Assume $\boldsymbol{w}_t$ is already known and let $\Delta \boldsymbol{w} = \boldsymbol{w}_{t+1} - \boldsymbol{w}_t$, we have

$$\underset{\boldsymbol{w}_{t+1} \in \mathbb{R}^n}{\arg \min} f(\boldsymbol{w}_{t+1}) = \underset{\Delta \boldsymbol{w} \in \mathbb{R}^n}{\arg \min} f(\Delta \boldsymbol{w} + \boldsymbol{w}_t)$$

$$\approx \underset{\Delta \boldsymbol{w} \in \mathbb{R}^n}{\arg \min} f(\boldsymbol{w}_t) + (\Delta \boldsymbol{w})^T \nabla f(\boldsymbol{w}_t) + \frac{1}{2}(\Delta \boldsymbol{w})^T \nabla^2 f(\boldsymbol{w}_t) \Delta \boldsymbol{w} \tag{2}$$

$$\approx \underset{\Delta \boldsymbol{w} \in \mathbb{R}^n}{\arg \min} f(\boldsymbol{w}_t) + (\Delta \boldsymbol{w})^T \boldsymbol{m}_t + \frac{1}{2\alpha_t}(\Delta \boldsymbol{w})^T B_t \Delta \boldsymbol{w},$$

where the first approximation is from Taylor expansion, and the second approximation are from $\boldsymbol{m}_t \approx \nabla f(\boldsymbol{w}_t)$ ($\boldsymbol{m}_t$ denotes the weighted average of gradient $\boldsymbol{g}_t$) and $\alpha_t \approx \frac{(\Delta \boldsymbol{w})^T B_t \Delta \boldsymbol{w}}{(\Delta \boldsymbol{w})^T \nabla^2 f(\boldsymbol{w}_t) \Delta \boldsymbol{w}}$ ($\alpha_t$ denotes the step size). By solving Equation (2), the general update formula is

$$\boldsymbol{w}_{t+1} = \boldsymbol{w}_t - \alpha_t B_t^{-1} \boldsymbol{m}_t, \quad t \in \{1, 2, \ldots, T\}, \tag{3}$$

where $B_t$ is the so-called preconditioning matrix that adjusts updated velocity of variable $\boldsymbol{w}_t$ in each direction. The majority of gradient descent algorithms can be succinctly encapsulated by Equation (3), ranging from the conventional second order optimizer, Gauss-Newton method, to the standard first-order optimizer, SGD, via different combinations of $B_t$ and $\boldsymbol{m}_t$. Table 1 summarizes different implementations of popular optimizers.

Intuitively, the closer $B_t$ approximates the Hessian, the faster convergence rate the optimizer can achieve in terms of number of iterations, since the Gauss-Hessian method enjoys a quadratic rate, whereas the gradient descent converges linearly under certain conditions (Theorems 1.2.4, 1.2.5 in Nesterov [25]). However, computing the Hessian is computationally expensive for large models. Thus, it is essential to strike a balance between the degree of Hessian approximation and computational efficiency when designing the preconditioning matrix.

Table 1: Different optimizers by choosing different $B_t$.

| $B_t$ | Optimizer |
|---|---|
| $B_t = \mathbf{H}$ | GAUSS-HESSIAN |
| $B_t \approx \mathbf{H}$ | BFGS [4, 13, 14, 31], LBFGS [5] |
| $B_t \approx \mathrm{diag}(\mathbf{H})$ | ADAHESSIAN [34] |
| $B_t = \mathbf{F}$ | NATURAL GRADIENT [2] |
| $B_t^2 \approx \mathbf{F}_{emp}$ | SHAMPOO [16] |
| $B_t^2 \approx \mathrm{diag}(\mathbf{F}_{emp})$ | ADAGRAD [12], ADADELTA [35], ADAM [18], ADAMW [21], AMSGRAD [28] |
| $B_t^2 \approx \mathrm{diag}(\mathbf{Var}(\boldsymbol{g}_t))$ | ADABELIEF [39] |
| $B_t = \mathbb{I}$ | SGD [29], MOMENTUM [27] |

$\mathbf{H}$ is the Hessian. $\mathbf{F}$ is the Fisher information matrix. $\mathbf{F}_{emp}$ is the empirical Fisher information matrix.

In this paper, we propose the AGD (**A**uto-switchable optimizer with **G**radient **D**ifference of adjacent steps) optimizer based on the idea of efficiently and effectively acquiring the information of the Hessian. The diagonal entries of AGD's preconditioning matrix are computed as the difference of gradients between two successive iterations, serving as an approximation of the inner product between the Hessian row vectors and difference of parameter vectors. In addition, AGD is equipped with an adaptive switching mechanism that automatically toggles its preconditioning matrix between SGD and the adaptive optimizer, governed by a threshold hyperparameter $\delta$ which enables AGD adaptive to various scenarios. Our contributions can be summarized as follows.

- We present a novel optimizer called AGD, which efficiently and effectively integrates the information of the Hessian into the preconditioning matrix and switches dynamically between SGD and the adaptive optimizer. We establish theoretical results of convergence guarantees for both non-convex and convex stochastic settings.

- We validate AGD on six public datasets: two from NLP (IWSLT14 [6] and PTB [23]), two from CV (Cifar10 [19] and ImageNet [30]), and the rest two from RecSys (Criteo [11] and Avazu [3]). The experimental results suggest that AGD is on par with or outperforms the SOTA optimizers.

- We analyze how AGD is able to switch automatically between SGD and the adaptive optimizer, and assess the effect of hyperparameter $\delta$ which controls the auto-switch process in different scenarios.

**Notation**

We use lowercase letters to denote scalars, boldface lowercase to denote vectors, and uppercase letters to denote matrices. We employ subscripts to denote a sequence of vectors, e.g., $\boldsymbol{x}_1, \ldots, \boldsymbol{x}_t$ where $t \in [T] := \{1, 2, \ldots, T\}$, and one additional subscript is used for specific entry of a vector, e.g., $x_{t,i}$ denotes $i$-th element of $\boldsymbol{x}_t$. For any vectors $\boldsymbol{x}, \boldsymbol{y} \in \mathbb{R}^n$, we write $\boldsymbol{x}^T \boldsymbol{y}$ or $\boldsymbol{x} \cdot \boldsymbol{y}$ for the standard inner product, $\boldsymbol{x}\boldsymbol{y}$ for element-wise multiplication, $\boldsymbol{x}/\boldsymbol{y}$ for element-wise division, $\sqrt{\boldsymbol{x}}$ for element-wise square root, $\boldsymbol{x}^2$ for element-wise square, and $\max(\boldsymbol{x}, \boldsymbol{y})$ for element-wise maximum. Similarly, any operator performed between a vector $\boldsymbol{x} \in \mathbb{R}^n$ and a scalar $c \in \mathbb{R}$, such as $\max(\boldsymbol{x}, c)$, is also element-wise. We denote $\|\boldsymbol{x}\| = \|\boldsymbol{x}\|_2 = \sqrt{\langle \boldsymbol{x}, \boldsymbol{x} \rangle}$ for the standard Euclidean norm, $\|\boldsymbol{x}\|_1 = \sum_i |x_i|$ for the $\ell_1$ norm, and $\|\boldsymbol{x}\|_\infty = \max_i |x_i|$ for the $\ell_\infty$-norm, where $x_i$ is the $i$-th element of $\boldsymbol{x}$.

Let $f_t(\boldsymbol{w})$ be the loss function of the model at $t$-step where $\boldsymbol{w} \in \mathbb{R}^n$. We consider $\boldsymbol{m}_t$ as Exponential Moving Averages (EMA) of $\boldsymbol{g}_t$ throughout this paper, i.e.,

$$\boldsymbol{m}_t = \beta_1 \boldsymbol{m}_{t-1} + (1 - \beta_1)\boldsymbol{g}_t = (1 - \beta_1) \sum_{i=1}^t \boldsymbol{g}_{t-i+1}\beta_1^{i-1}, \ t \geq 1, \tag{4}$$

where $\beta_1 \in [0, 1)$ is the exponential decay rate.

## 2    Related work

ASGD [36] leverages Taylor expansion to estimate the gradient at the global step in situations where the local worker's gradient is delayed, by analyzing the relationship between the gradient difference and Hessian. To approximate the Hessian, the authors utilize the diagonal elements of empirical Fisher information due to the high computational and spatial overhead of Hessian. ADABELIEF [39] employs the EMA of the gradient as the predicted gradient and adapts the step size by scaling it with the difference between predicted and observed gradients, which can be considered as the variance of the gradient.

Hybrid optimization methods, including ADABOUND [22] and SWATS [17], have been proposed to enhance generalization performance by switching an adaptive optimizer to SGD. ADABOUND utilizes learning rate clipping on ADAM, with upper and lower bounds that are non-increasing and non-decreasing functions, respectively. One can show that it ultimately converges to the learning rate of SGD. Similarly, SWATS also employs the clipping method, but with constant upper and lower bounds.

## 3    Algorithm

### 3.1    Details of AGD optimizer

---

**Algorithm 1** AGD

1:  **Input:** parameters $\beta_1$, $\beta_2$, $\delta$, $\boldsymbol{w}_1 \in \mathbb{R}^n$, step size $\alpha_t$, initialize $\boldsymbol{m}_0 = \boldsymbol{0}, \boldsymbol{b}_0 = \boldsymbol{0}$

2:  **for** $t = 1$ **to** $T$ **do**

3:      $\boldsymbol{g}_t = \nabla f_t(\boldsymbol{w}_t)$

4:      $\boldsymbol{m}_t \leftarrow \beta_1 \boldsymbol{m}_{t-1} + (1 - \beta_1)\boldsymbol{g}_t$

5:      $\boldsymbol{s}_t = \begin{cases} \frac{\boldsymbol{m}_1}{1-\beta_1} & t = 1 \\ \frac{\boldsymbol{m}_t}{1-\beta_1^t} - \frac{\boldsymbol{m}_{t-1}}{1-\beta_1^{t-1}} & t > 1 \end{cases}$

6:      $\boldsymbol{b}_t \leftarrow \beta_2 \boldsymbol{b}_{t-1} + (1 - \beta_2)\boldsymbol{s}_t^2$

7:      $\boldsymbol{w}_{t+1} = \boldsymbol{w}_t - \alpha_t \frac{\sqrt{1-\beta_2^t}}{1-\beta_1^t} \frac{\boldsymbol{m}_t}{\max(\sqrt{\boldsymbol{b}_t}, \delta\sqrt{1-\beta_2^t})}$

8:  **end for**

---

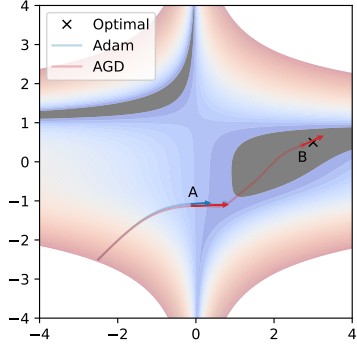

Figure 1: Trajectories of AGD and Adam in the Beale function.

Algorithm 1 summarizes our AGD algorithm. The design of AGD comes from two parts: gradient difference and auto switch for faster convergence and better generalization performance across tasks.

**Gradient difference**   Our motivation stems from how to efficiently and effectively integrate the information of the Hessian into the preconditioning matrix. Let $\Delta \boldsymbol{w} = \boldsymbol{w}_t - \boldsymbol{w}_{t-1}$ and $\nabla_i f$ denote the $i$-th element of $\nabla f$. From Taylor expansion or the mean value theorem, when $\|\Delta \boldsymbol{w}\|$ is small, we have the following approximation,

$$\nabla_i f(\boldsymbol{w}_t) - \nabla_i f(\boldsymbol{w}_{t-1}) \approx \nabla \nabla_i f(\boldsymbol{w}_t) \cdot \Delta \boldsymbol{w}.$$

It means the difference of gradients between adjacent steps can be an approximation of the inner product between the Hessian row vectors and the difference of two successive parameter vectors. To illustrate the effectiveness of gradient difference in utilizing Hessian information, we compare the convergence trajectories of AGD and Adam on the Beale function. As shown in Figure 1, we see that AGD converges much faster than Adam; when AGD reaches the optimal point, Adam has only covered about half of the distance. We select the two most representative points on the AGD trajectory in the figure, the maximum and minimum points of $\|\nabla f\|_1 / \|\text{diag}(H)\|_1$, to illustrate how AGD accelerates convergence by utilizing Hessian information. At the maximum point (A), where the gradient is relatively large and the curvature is relatively small ($\|\nabla f\|_1 = 22.3$, $\|\text{diag}(H)\|_1 = 25.3$), the step size of AGD is 1.89 times that of Adam. At the minimum point (B), where the gradient is relatively small and the curvature is relatively large ($\|\nabla f\|_1 = 0.2$, $\|\text{diag}(H)\|_1 = 34.8$), the step size decreases to prevent it from missing the optimal point during the final convergence phase.

To approximate $\nabla f(\boldsymbol{w}_t)$, we utilize $\boldsymbol{m}_t / (1 - \beta_1^t)$ instead of $\boldsymbol{g}_t$, as the former provides an unbiased estimation of $\nabla f(\boldsymbol{w}_t)$ with lower variance. According to Kingma and Ba [18], we have $\mathbf{E}\left[\frac{\boldsymbol{m}_t}{1-\beta_1^t}\right] \approx \mathbf{E}[\boldsymbol{g}_t]$, where the equality is satisfied if $\{\boldsymbol{g}_t\}$ is stationary. Additionally, assuming $\{\boldsymbol{g}_t\}$ is strictly stationary and $\mathbf{Cov}(\boldsymbol{g}_i, \boldsymbol{g}_j) = 0$ if $i \neq j$ for simplicity and $\beta_1 \in (0, 1)$, we observe that

$$\mathbf{Var}\left[\frac{\boldsymbol{m}_t}{1-\beta_1^t}\right] = \frac{1}{(1-\beta_1^t)^2}\mathbf{Var}\left[(1-\beta_1)\sum_{i=1}^{t}\beta_1^{t-i}\boldsymbol{g}_i\right] = \frac{(1+\beta_1^t)(1-\beta_1)}{(1-\beta_1^t)(1+\beta_1)}\mathbf{Var}[\boldsymbol{g}_t] < \mathbf{Var}[\boldsymbol{g}_t].$$

Now, we denote

$$\boldsymbol{s}_t = \begin{cases} \boldsymbol{m}_1 / (1 - \beta_1) & t = 1, \\ \boldsymbol{m}_t / (1 - \beta_1^t) - \boldsymbol{m}_{t-1} / (1 - \beta_1^{t-1}) & t > 1, \end{cases}$$

and design the preconditioning matrix $B_t$ satisfying

$$B_t^2 = \text{diag}(\text{EMA}(\boldsymbol{s}_1\boldsymbol{s}_1^T, \boldsymbol{s}_2\boldsymbol{s}_2^T, \cdots, \boldsymbol{s}_t\boldsymbol{s}_t^T)) / (1 - \beta_2^t),$$

where $\beta_2$ represents the parameter of EMA and bias correction is achieved via the denominator.

Note that previous research, such as the one discussed in Section 2 by Zheng et al. [36], has acknowledged the correlation between the difference of two adjacent gradients and the Hessian. However, the key difference is that they did not employ this relationship to construct an optimizer. In contrast, our approach presented in this paper leverages this relationship to develop an optimizer, and its effectiveness has been validated in the experiments detailed in Section 4.

**Auto switch**   Typically a small value is added to $B_t$ for numerical stability, resulting in $B_t + \epsilon \mathbb{I}$. However, in this work we propose to replace this with $\max(B_t, \delta\mathbb{I})$, where we use a different notation $\delta$ to emphasize its crucial role in auto-switch mechanism. In contrast to $\epsilon$, $\delta$ can be a relatively large (such as 1e-2). If the element of $\hat{\boldsymbol{b}}_t := \sqrt{\boldsymbol{b}_t / (1 - \beta_2^t)}$ exceeds $\delta$, AGD (Line 7 of Algorithm 1) takes a confident adaptive step. Otherwise, the update is performed using EMA, i.e., $\boldsymbol{m}_t$, with a constant scale of $\alpha_t / (1 - \beta_1^t)$, similar to SGD with momentum. It's worth noting that, AGD can automatically switch modes on a per-parameter basis as the training progresses.

Compared to the commonly used additive method, AGD effectively eliminates the noise generated by $\epsilon$ during adaptive updates. In addition, AGD offers an inherent advantage of being able to generalize across different tasks by tuning the value of $\delta$, obviating the need for empirical choices among a plethora of optimizers.

## 3.2 Comparison with other optimizers

**Comparison with AdaBound** As noted in Section 2, the auto-switch bears similarities to AdaBound [22] in its objective to enhance the generalization performance by switching to SGD using the clipping method. Nonetheless, the auto-switch's design differs significantly from AdaBound. Rather than relying solely on adaptive optimization in the early stages, AGD has the flexibility to switch seamlessly between stochastic and adaptive methods, as we will demonstrate in Section 4.5. In addition, AGD outperforms AdaBound's across various tasks, as we will show in Appendix A.2.

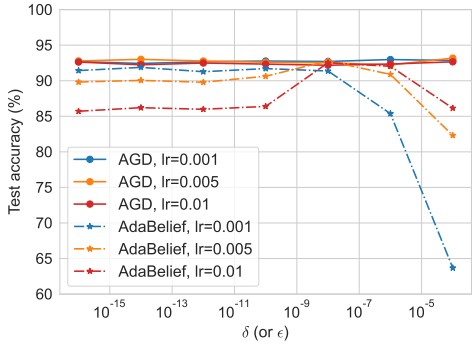

Figure 2: Comparison of stability between AGD and AdaBelief relative to the parameter $\delta$ (or $\epsilon$) for ResNet32 on Cifar10. AGD shows better stability over a wide range of $\delta$ (or $\epsilon$) variations than AdaBelief.

**Comparison with AdaBelief** While in principle our design is fundamentally different from that of AdaBelief [39], which approximates gradient variance with its preconditioning matrix, we do see some similarities in our final forms. Compared with the denominator of AGD, the denominator of AdaBelief $\boldsymbol{s}_t = \boldsymbol{g}_t - \boldsymbol{m}_t = \frac{\beta_1}{1-\beta_1}(\boldsymbol{m}_t - \boldsymbol{m}_{t-1})$ lacks bias correction for the subtracted terms and includes a multiplication factor of $\frac{\beta_1}{1-\beta_1}$. In addition, we observe that AGD exhibits superior stability compared to AdaBelief. As shown in Figure 2, when the value of $\epsilon$ deviates from 1e-8 by orders of magnitude, the performance of AdaBelief degrades significantly; in contrast, AGD maintains good stability over a wide range of $\delta$ variations.

## 3.3 Numerical analysis

In this section, we present a comparison between AGD and several SOTA optimizers on three test functions. We use the parameter settings from Zhuang et al. [39], where the learning rate is set to 1e-3 for all adaptive optimizers, along with the same default values of $\epsilon$ (or $\delta$) (1e-8) and betas ($\beta_1 = 0.9, \beta_2 = 0.999$). For SGD, we set the momentum to 0.9 and the learn-

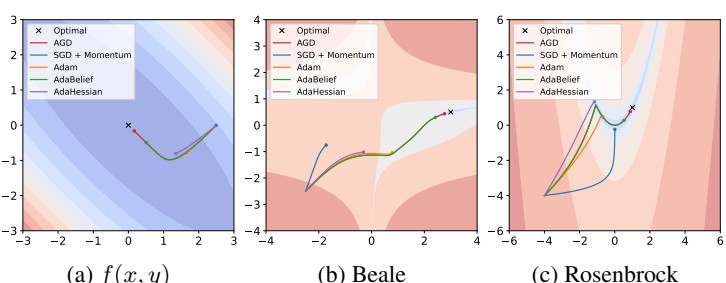

(a) $f(x, y)$      (b) Beale      (c) Rosenbrock

Figure 3: Trajectories of different optimizers in three test functions, where $f(x, y) = (x + y)^2 + (x - y)^2/10$. We also provide animated versions at https://youtu.be/Qv5X3v5YUw0.

ing rate to 1e-6 to ensure numerical stability. As shown in Figure 3, AGD exhibits promising results by firstly reaching the optimal points in all experiments before other competitors. In addition, we conduct a search of the largest learning rate for each optimizer with respect to the Beale function, and AGD once again stands out as the strongest method. More details on the search process can be found in Appendix A.3.

## 4 Experiments

### 4.1 Experiment setup

We extensively compared the performance of various optimizers on diverse learning tasks in NLP, CV, and RecSys; we only vary the settings for the optimizers and keep the other settings consistent in this evaluation. To offer a comprehensive analysis, we provide a detailed description of each task and the optimizers' efficacy in different application domains.

**NLP:** We conduct experiments using Language Modeling (LM) on Penn TreeBank [23] and Neural Machine Translation (NMT) on IWSLT14 German-to-English (De-En) [6] datasets. For the LM task,

we train 1, 2, and 3-layer LSTM models with a batch size of 20 for 200 epochs. For the NMT task, we implement the Transformer `small` architecture, and employ the same pre-processing method and settings as AdaHessian [34], including a length penalty of 1.0, beam size of 5, and max tokens of 4096. We train the model for 55 epochs and average the last 5 checkpoints for inference. We maintain consistency in our learning rate scheduler and warm-up steps. Table 2 provides complete details of the experimental setup.

**CV:** We conduct experiments using ResNet20 and ResNet32 on the Cifar10 [19] dataset, and ResNet18 on the ImageNet [30] dataset, as detailed in Table 2. It is worth noting that the number of parameters of ResNet18 is significantly larger

Table 2: Experiments setup.

| Task | Dataset | Model | Train | Val/Test | Params |
|------|---------|-------|-------|----------|--------|
| NLP-LM | PTB | 1-layer LSTM
2-layer LSTM
3-layer LSTM | 0.93M | 730K/82K | 5.3M
13.6M
24.2M |
| NLP-NMT | IWSLT14 De-En | Transformer `small` | 153K | 7K/7K | 36.7M |
| CV | Cifar10
ImageNet | ResNet20/ResNet32
ResNet18 | 50K
1.28M | 10K
50K | 0.27M/0.47M
11.69M |
| RecSys | Avazu
Criteo | MLP
DCN | 36.2M
39.4M | 4.2M
6.6M | 151M
270M |

than that of ResNet20/32, stemming from inconsistencies in ResNet's naming conventions. Within the ResNet architecture, the consistency in filter sizes, feature maps, and blocks is maintained only within specific datasets. Originally proposed for ImageNet, ResNet18 is more complex compared to ResNet20 and ResNet32, which were tailored for the less demanding Cifar10 dataset. Our training process involves 160 epochs with a learning rate decay at epochs 80 and 120 by a factor of 10 for Cifar10, and 90 epochs with a learning rate decay every 30 epochs by a factor of 10 for ImageNet. The batch size for both datasets is set to 256.

**RecSys:** We conduct experiments on two widely used datasets, Avazu [3] and Criteo [11], which contain logs of display ads. The goal is to predict the Click Through Rate (CTR). We use the samples from the first nine days of Avazu for training and the remaining samples for testing. We employ the Multilayer Perceptron (MLP) structure (a fundamental architecture used in most deep CTR models). The model maps each categorical feature into a 16-dimensional embedding vector, followed by four fully connected layers of dimensions 64, 32, 16, and 1, respectively. For Criteo, we use the first 6/7 of all samples as the training set and last 1/7 as the test set. We adopt the Deep & Cross Network (DCN) [32] with an embedding size of 8, along with two deep layers of size 64 and two cross layers. Detailed summary of the specifications can be found in Table 2. For both datasets, we train them for one epoch using a batch size of 512.

Optimizers to compare include SGD [29], Adam [18], AdamW [21], AdaBelief [39] and AdaHessian [34]. To determine each optimizer's hyperparameters, we adopt the parameters suggested in the literature of AdaHessian and AdaBelief when the experimental settings are identical. Otherwise, we perform hyperparameter searches for optimal settings. A detailed description of this process can be found in Appendix A.1. For our NLP and CV experiments, we utilize GPUs with the PyTorch framework [26], while our RecSys experiments are conducted with three parameter servers and five workers in the TensorFlow framework [1]. To ensure the reliability of our results, we execute each experiment five times with different random seeds and calculate statistical results.

## 4.2 NLP

We report the perplexity (PPL, lower is better) and case-insensitive BiLingual Evaluation Understudy (BLEU, higher is better) score on test set for LM and NMT tasks, respectively. The results are shown in Table 3. For the LM task on PTB, AGD achieves the lowest PPL in all 1,2,3-layer LSTM experiments, as demonstrated in Figure 4. For the NMT task on IWSLT14, AGD is on par with AdaBelief, but outperforms the other optimizers.

## 4.3 CV

Table 4 reports the top-1 accuracy for different optimizers when trained on Cifar10 and ImageNet. It is remarkable that AGD outperforms other optimizers on both Cifar10 and ImageNet. The test accuracy ($[\mu \pm \sigma]$) curves of different optimizers for ResNet20/32 on Cifar10 and ResNet18 on

Table 3: Test PPL and BLEU score for LM and NMT tasks. † is reported in AdaHessian [34].

| Dataset | | PTB | | IWSLT14 |
| Metric | | PPL, lower is better | | BLEU, higher is better |
| Model | 1-layer LSTM | 2-layer LSTM | 3-layer LSTM | Transformer |
|---|---|---|---|---|
| SGD | $85.36 \pm .34$ $(-4.13)$ | $67.26 \pm .17$ $(-1.42)$ | $63.68 \pm .17$ $(-2.79)$ | $28.57 \pm .15$†$(+7.37)$ |
| Adam | $84.50 \pm .16$ $(-3.27)$ | $67.01 \pm .11$ $(-1.17)$ | $64.45 \pm .26$ $(-3.56)$ | $32.93 \pm .26$ $(+3.01)$ |
| AdamW | $88.16 \pm .19$ $(-6.93)$ | $95.25 \pm 1.33$ $(-29.41)$ | $102.61 \pm 1.13$ $(-41.72)$ | $35.82 \pm .06$ $(+0.12)$ |
| AdaBelief | $84.40 \pm .21$ $(-3.17)$ | $66.69 \pm .23$ $(-0.85)$ | $61.34 \pm .11$ $(-0.45)$ | $35.93 \pm .08$ $(+0.01)$ |
| AdaHessian | $88.62 \pm .15$ $(-7.39)$ | $73.37 \pm .22$ $(-7.53)$ | $69.51 \pm .19$ $(-8.62)$ | $35.79 \pm .06$†$(+0.15)$ |
| **AGD** | $\mathbf{81.23 \pm .17}$ | $\mathbf{65.84 \pm .18}$ | $\mathbf{60.89 \pm .09}$ | $\mathbf{35.94 \pm .11}$ |

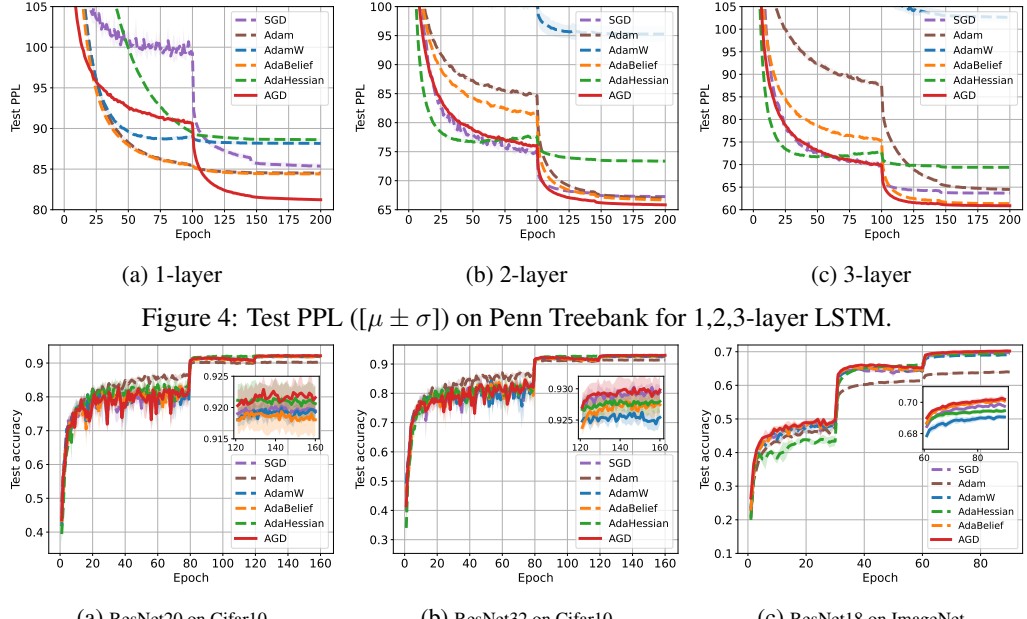

(a) 1-layer      (b) 2-layer      (c) 3-layer

Figure 4: Test PPL ($[\mu \pm \sigma]$) on Penn Treebank for 1,2,3-layer LSTM.

(a) ResNet20 on Cifar10      (b) ResNet32 on Cifar10      (c) ResNet18 on ImageNet

Figure 5: Test accuracy ($[\mu \pm \sigma]$) of different optimizers for ResNet20/32 on Cifar10 and ResNet18 on ImageNet.

ImageNet are illustrated in Figure 5. Notice that the numbers of SGD and AdaHessian on ImageNet are lower than the numbers reported in original papers [7, 34], which were run only once (we average multiple trials here). AdaHessian can achieve $70.08\%$ top-1 accuracy in Yao et al. [34] while we report $69.57 \pm 0.12\%$. Due to the limited training details provided in Yao et al. [34], it is difficult for us to explain the discrepancy. However, regardless of which result of AdaHessian is taken, AGD outperforms AdaHessian significantly. Our reported top-1 accuracy of SGD is $69.94 \pm 0.10\%$, slightly lower than $70.23\%$ reported in Chen et al. [7]. We find that the differences in training epochs, learning rate scheduler and weight decay rate are the main reasons. We also run the experiment using the same configuration as in Chen et al. [7], and AGD can achieve $70.45\%$ accuracy at lr = 4e-4 and $\delta$ = 1e-5, which is still better than the $70.23\%$ result reported in Chen et al. [7].

We also report the accuracy of AGD for ResNet18 on Cifar10 for comparing with the SOTA results [3], which is listed in Appendix A.5. Here we clarify again the ResNet naming confusion. The test accuracy of ResNet18 on Cifar10 training with AGD is above 95%, while ResNet32 is about 93% since ResNet18 is much more complex than ResNet32.

## 4.4 RecSys

To evaluate the accuracy of CTR estimation, we have adopted the Area Under the receiver-operator Curve (AUC) as our evaluation criterion, which is widely recognized as a reliable measure [15]. As stated in Cheng et al. [10], Wang et al. [32], Ling et al. [20], Zhu et al. [38], even an absolute

---

[3]https://paperswithcode.com/sota/stochastic-optimization-on-cifar-10-resnet-18

Table 4: Top-1 accuracy for different optimizers when trained on Cifar10 and ImageNet.

| Dataset | Cifar10 | | ImageNet |
|---|---|---|---|
| Model | ResNet20 | ResNet32 | ResNet18 |
| SGD | 92.14 ± .14 (+0.21) | 93.10 ± .07 (+0.02) | 69.94 ± .10 (+0.41) |
| Adam | 90.46 ± .20 (+1.89) | 91.54 ± .12 (+1.58) | 64.03 ± .16 (+6.32) |
| AdamW | 92.12 ± .14 (+0.23) | 92.72 ± .01 (+0.40) | 69.11 ± .17 (+1.24) |
| AdaBelief | 92.19 ± .15 (+0.16) | 92.90 ± .13 (+0.22) | 70.20 ± .03 (+0.15) |
| AdaHessian | 92.27 ± .27 (+0.08) | 92.91 ± .14 (+0.21) | 69.57 ± .12 (+0.78) |
| AGD | **92.35 ± .24** | **93.12 ± .18** | **70.35 ± .17** |

Table 5: Test AUC for different optimizers when trained on Avazu and Criteo.

| Dataset | Avazu | Criteo |
|---|---|---|
| Model | MLP | DCN |
| SGD | 0.7463 ± .0005 (+1.7‰) | 0.7296 ± .0067 (+72.7‰) |
| Adam | 0.7458 ± .0010 (+2.2‰) | **0.8023 ± .0002** (+0.0‰) |
| AdaBelief | 0.7467 ± .0009 (+1.3‰) | 0.8022 ± .0002 (+0.1‰) |
| AdaHessian | 0.7434 ± .0006 (+4.6‰) | 0.8004 ± .0005 (+1.9‰) |
| AGD | **0.7480 ± .0008** | **0.8023 ± .0004** |

improvement of 1‰ in AUC can be considered practically significant given the difficulty of improving CTR prediction. Our experimental results in Table 5 indicate that AGD can achieve highly competitive or significantly better performance when compared to other optimizers. In particular, on the Avazu task, AGD outperforms all other optimizers by more than 1‰. On the Criteo task, AGD performs better than SGD and AdaHessian, and achieves comparable performance to Adam and AdaBelief.

## 4.5 The effect of $\delta$

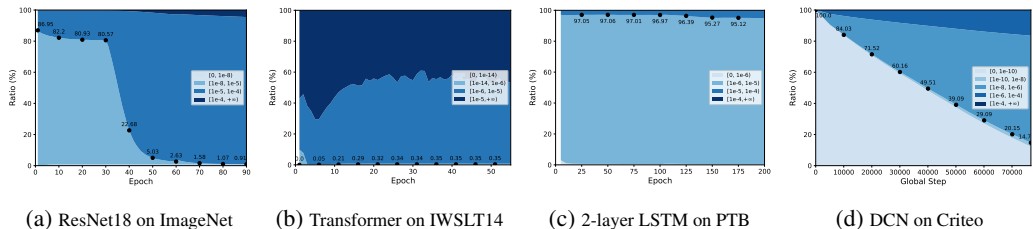

(a) ResNet18 on ImageNet  (b) Transformer on IWSLT14  (c) 2-layer LSTM on PTB  (d) DCN on Criteo

Figure 6: The distribution of $\hat{b}_t$ on different epochs/steps. The colored area denotes the ratio of $\hat{b}_t$ in the corresponding interval. The values of $\delta$ for ResNet18 on ImageNet, Transformer on IWSLT14, 2-layer LSTM on PTB, and DCN on Criteo are 1e-5, 1e-14, 1e-5 and 1e-8, respectively.

In this section, we aim to provide a comprehensive analysis of the impact of $\delta$ on the training process by precisely determining the percentage of $\hat{b}_t$ that Algorithm 1 truncates. To this end, Figure 6 shows the distribution of $\hat{b}_t$ across various tasks under the optimal configuration that we have identified. The black dot on the figure provides the precise percentage of $\hat{b}_t$ that $\delta$ truncates during the training process. Notably, a lower percentage indicates a higher degree of SGD-like updates compared to adaptive steps, which can be adjusted through $\delta$.

As SGD with momentum generally outperforms adaptive optimizers on CNN tasks [34, 39], we confirm this observation as illustrated in Figure 6a: AGD behaves more like SGD during the initial stages of training (before the first learning rate decay at the 30th epoch) and switches to adaptive optimization for fine-tuning. Figure 6b indicates that the parameters taking adaptive updates are dominant, as expected because adaptive optimizers such as AdamW are preferred in transformers. Figure 6c demonstrates that most parameters update stochastically, which explains why AGD has a similar curve to SGD in Figure 4b before the 100th epoch. The proportion of parameters taking adaptive updates grows from 3% to 5% afterward, resulting in a better PPL in the fine-tuning stage. Concerning Figure 6d, the model of the RecSys task trains for only one epoch, and AGD gradually switches to adaptive updates for a better fit to the data.

## 4.6 Computational cost

We train a Transformer `small` model for IWSLT14 on a single NVIDIA P100 GPU. AGD is comparable to the widely used AdamW optimizer, while significantly outperforms AdaHessian in terms of memory footprint and training speed. As a result, AGD can be a drop-in replacement for AdamW with similar computation cost and better generalization performance.

Table 6: Computational cost for Transformer `small`.

| Optimizer | Memory | Time per Epoch | Relative time to AdamW |
|-----------|--------|----------------|------------------------|
| SGD | 5119 MB | 230 s | 0.88× |
| AdamW | 5413 MB | 260 s | 1.00× |
| AdaHessian | 8943 MB | 750 s | 2.88× |
| AGD | 5409 MB | 278 s | 1.07× |

## 5 Theoretical analysis

Using the framework developed in Reddi et al. [28], Yang et al. [33], Chen et al. [9], Zhou et al. [37], we have the following theorems that provide the convergence in non-convex and convex settings. Particularly, we use $\beta_{1,t}$ to replace $\beta_1$, where $\beta_{1,t}$ is non-increasing with respect to $t$.

**Theorem 1.** *(Convergence in non-convex settings) Suppose that the following assumptions are satisfied:*

1. *$f$ is differential and lower bounded, i.e., $f(\boldsymbol{w}^*) > -\infty$ where $\boldsymbol{w}^*$ is an optimal solution. $f$ is also L-smooth, i.e., $\forall \boldsymbol{u}, \boldsymbol{v} \in \mathbb{R}^n$, we have $f(\boldsymbol{u}) \leq f(\boldsymbol{v}) + \langle \nabla f(\boldsymbol{v}), \boldsymbol{u} - \boldsymbol{v} \rangle + \frac{L}{2}\|\boldsymbol{u} - \boldsymbol{v}\|^2$.*

2. *At step t, the algorithm can access a bounded noisy gradient and the true gradient is bounded, i.e., $\|\boldsymbol{g}_t\|_\infty \leq G_\infty, \|\nabla f(\boldsymbol{w}_t)\|_\infty \leq G_\infty, \forall t \in [T]$. Without loss of generality, we assume $G_\infty \geq \delta$.*

3. *The noisy gradient is unbiased and the noise is independent, i.e., $\boldsymbol{g}_t = \nabla f(\boldsymbol{w}_t) + \boldsymbol{\zeta}_t, \mathbf{E}[\boldsymbol{\zeta}_t] = \mathbf{0}$ and $\boldsymbol{\zeta}_i$ is independent of $\boldsymbol{\zeta}_j$ if $i \neq j$.*

4. *$\alpha_t = \alpha/\sqrt{t}$, $\beta_{1,t}$ is non-increasing satisfying $\beta_{1,t} \leq \beta_1 \in [0, 1)$, $\beta_2 \in [0, 1)$ and $b_{t,i} \leq b_{t+1,i}\ \forall i \in [n]$.*

*Then Algorithm 1 yields*

$$\min_{t \in [T]} \mathbf{E}[\|\nabla f(\boldsymbol{w}_t)\|^2] < C_3 \frac{1}{\sqrt{T} - \sqrt{2}} + C_4 \frac{\log T}{\sqrt{T} - \sqrt{2}} + C_5 \frac{\sum_{t=1}^T \hat{\alpha}_t(\beta_{1,t} - \beta_{1,t+1})}{\sqrt{T} - \sqrt{2}}, \quad (5)$$

*where $C_3$, $C_4$ and $C_5$ are defined as follows:*

$$C_3 = \frac{G_\infty}{\alpha(1 - \beta_1)^2(1 - \beta_2)^2}\left(f(\boldsymbol{w}_1) - f(\boldsymbol{w}^*) + \frac{nG_\infty^2\alpha}{(1 - \beta_1)^8\delta^2}(\delta + 8L\alpha) + \frac{\alpha\beta_1 nG_\infty^2}{(1 - \beta_1)^3\delta}\right),$$

$$C_4 = \frac{15LnG_\infty^3\alpha}{2(1 - \beta_2)^2(1 - \beta_1)^{10}\delta^2}, \quad C_5 = \frac{nG_\infty^3}{\alpha(1 - \beta_1)^5(1 - \beta_2)^2\delta}.$$

The proof of Theorem 1 is presented in Appendix B. There are two important points that should be noted: Firstly, in assumption 2, we can employ the gradient norm clipping technique to ensure the upper bound of the gradients. Secondly, in assumption 4, $b_{t,i} \leq b_{t+1,i}\ \forall i \in [n]$, which is necessary for the validity of Theorems 1 and 2, may not always hold. To address this issue, we can implement the AMSGrad condition [28] by setting $\boldsymbol{b}_{t+1} = \max(\boldsymbol{b}_{t+1}, \boldsymbol{b}_t)$. However, this may lead to a potential decrease in the algorithm's performance in practice. The more detailed analysis is provided in Appendix A.4. From Theorem 1, we have the following corollaries.

**Corollary 1.** *Suppose $\beta_{1,t} = \beta_1/\sqrt{t}$, we have*

$$\min_{t \in [T]} \mathbf{E}[\|\nabla f(\boldsymbol{w}_t)\|^2] < C_3 \frac{1}{\sqrt{T} - \sqrt{2}} + C_4 \frac{\log T}{\sqrt{T} - \sqrt{2}} + \frac{C_5\alpha}{1 - \beta_1}\frac{\log T + 1}{\sqrt{T} - \sqrt{2}},$$

*where $C_3$, $C_4$ and $C_5$ are the same with Theorem 1.*

The proof of Corollary 1 can be found in Appendix C.

**Corollary 2.** *Suppose $\beta_{1,t} = \beta_1, \forall t \in [T]$, we have*

$$\min_{t \in [T]} \mathbf{E}[\|\nabla f(\boldsymbol{w}_t)\|^2] < C_3 \frac{1}{\sqrt{T} - \sqrt{2}} + C_4 \frac{\log T}{\sqrt{T} - \sqrt{2}},$$

*where $C_3$ and $C_4$ are the same with Theorem 1.*

Corollaries 1 and 2 imply the convergence (to the stationary point) rate for AGD is $O(\log T/\sqrt{T})$ in non-convex settings.

**Theorem 2.** *(Convergence in convex settings) Let $\{w_t\}$ be the sequence obtained by AGD (Algorithm 1), $\alpha_t = \alpha/\sqrt{t}$, $\beta_{1,t}$ is non-increasing satisfying $\beta_{1,t} \leq \beta_1 \in [0,1)$, $\beta_2 \in [0,1)$, $b_{t,i} \leq b_{t+1,i} \, \forall i \in [n]$ and $\|g_t\|_\infty \leq G_\infty, \forall t \in [T]$. Suppose $f_t(w)$ is convex for all $t \in [T]$, $w^*$ is an optimal solution of $\sum_{t=1}^T f_t(w)$, i.e., $w^* = \arg\min_{w \in \mathbb{R}^n} \sum_{t=1}^T f_t(w)$ and there exists the constant $D_\infty$ such that $\max_{t \in [T]} \|w_t - w^*\|_\infty \leq D_\infty$. Then we have the following bound on the regret*

$$\sum_{t=1}^T (f_t(w_t) - f_t(w^*)) < \frac{1}{1-\beta_1}\left(C_1\sqrt{T} + \sum_{t=1}^T \frac{\beta_{1,t}}{2\hat{\alpha}_t}nD_\infty^2 + C_2\sqrt{T}\right),$$

*where $C_1$ and $C_2$ are defined as follows:*

$$C_1 = \frac{n(2G_\infty + \delta)D_\infty^2}{2\alpha\sqrt{1-\beta_2}(1-\beta_1)^2}, \quad C_2 = \frac{n\alpha G_\infty^2}{(1-\beta_1)^3}\left(1 + \frac{1}{\delta\sqrt{1-\beta_2}}\right).$$

The proof of Theorem 2 is given in Appendix D. To ensure that the condition $\max_{t\in[T]}\|w_t - w^*\|_\infty \leq D_\infty$ holds, we can assume that the domain $\mathcal{W} \subseteq \mathbb{R}^n$ is bounded and project the sequence $\{w_t\}$ onto $\mathcal{W}$ by setting $w_{t+1} = \Pi_\mathcal{W}\left(w_t - \alpha_t \frac{\sqrt{1-\beta_2^t}}{1-\beta_1^t}\frac{m_t}{\max(\sqrt{b_t}, \delta\sqrt{1-\beta_2^t})}\right)$. From Theorem 2, we have the following corollary.

**Corollary 3.** *Suppose $\beta_{1,t} = \beta_1/t$, we have*

$$\sum_{t=1}^T (f_t(w_t) - f_t(w^*)) < \frac{1}{1-\beta_1}\left(C_1\sqrt{T} + \frac{nD_\infty^2\beta_1}{\alpha\sqrt{1-\beta_2}}\sqrt{T} + C_2\sqrt{T}\right),$$

*where $C_1$ and $C_2$ are the same with Theorem 2.*

The proof of Corollary 3 is given in Appendix E. Corollary 3 implies the regret is $O(\sqrt{T})$ and can achieve the convergence rate $O(1/\sqrt{T})$ in convex settings.

# 6 Conclusion

In this paper, we introduce a novel optimizer, AGD, which incorporates the Hessian information into the preconditioning matrix and allows seamless switching between SGD and the adaptive optimizer. We provide theoretical convergence rate proofs for both non-convex and convex stochastic settings and conduct extensive empirical evaluations on various real-world datasets. The results demonstrate that AGD outperforms other optimizers in most cases, resulting in significant performance improvements. Additionally, we analyze the mechanism that enables AGD to automatically switch between stochastic and adaptive optimization and investigate the impact of the hyperparameter $\delta$ for this process.

# Acknowledgement

We thank Yin Lou for his meticulous review of our manuscript and for offering numerous valuable suggestions.

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

# Appendix

## A  Details of experiments

### A.1  Configuration of optimizers

In this section, we provide a thorough description of the hyperparameters used for different optimizers across various tasks. For optimizers other than AGD, we adopt the recommended parameters for the identical experimental setup as indicated in the literature of AdaHessian [34] and AdaBelief [39]. In cases where these recommendations are not accessible, we perform a hyperparameter search to determine the optimal hyperparameters.

**NLP**

- SGD/Adam/AdamW: For the NMT task, we report the results of SGD from AdaHessian [34], and search learning rate among {5e-5, 1e-4, 5e-4, 1e-3} and epsilon in {1e-12, 1e-10, 1e-8, 1e-6, 1e-4} for Adam/AdamW, and for both optimizers the optimal learning rate/epsilon is 5e-4/1e-12. For the LM task, we follow the settings from AdaBelief [39], setting learning rate to 30 for SGD and 0.001 for Adam/AdamW while epsilon to 1e-12 for Adam/AdamW when training 1-layer LSTM. For 2-layer LSTM, we conduct a similar search with learning rate among {1e-4, 5e-4, 1e-3, 1e-2} and epsilon within {1e-12, 1e-10, 1e-8, 1e-6, 1e-4}, and the best parameters are learning rate = 1e-2 and epsilon = 1e-8/1e-4 for Adam/AdamW. For 3-layer LSTM, learning rate = 1e-2 and epsilon = 1e-8 are used for Adam/AdamW.

- AdaBelief: For the NMT task we use the recommended configuration from the latest implementation[4] for transformer. We search learning rate in {5e-5, 1e-4, 5e-4, 1e-3} and epsilon in {1e-16, 1e-14, 1e-12, 1e-10, 1e-8}, and the best configuration is to set learning rate as 5e-4 and epsilon as 1e-16. We adopt the same LSTM experimental setup for the LM task and reuse the optimal settings provided by AdaBelief [39], except for 2-layer LSTM, where we search for the optimial learning rate in {1e-4, 5e-4, 1e-3, 1e-2} and epsilon in {1e-16, 1e-14, 1e-12, 1e-10, 1e-8}. However, the best configuration is identical to the recommended.

- AdaHessian: For the NMT task, we adopt the same experimental setup as in the official implementation.[5] For LM task, we search the learning rate among {1e-3, 1e-2, 0.1, 1} and hessian power among {0.5, 1, 2}. We finally select 0.1 for learning rate and 0.5 for hessian power for 1-layer LSTM, and 1.0 for learning rate and 0.5 for for hessian power for 2,3-layer LSTM. Note that AdaHessian appears to overfit when using learning rate 1.0. Accordingly, we also try to decay its learning rate at the 50th/90th epoch, but it achieves a similar PPL.

- AGD: For the NMT task, we search learning rate among {5e-5, 1e-4, 5e-4, 1e-3} and $\delta$ among {1e-16, 1e-14, 1e-12, 1e-10, 1e-8}. We report the best result with learning rate 5e-5 and $\delta$ as 1e-14 for AGD. For the LM task, we search learning rate among {1e-4, 5e-4, 1e-3, 5e-3, 1e-2} and $\delta$ from 1e-16 to 1e-4, and the best settings for learning rate ($\delta$) is 5e-4 (1e-10) and 1e-3 (1e-5) for 1-layer LSTM (2,3-layer LSTM).

The weight decay is set to 1e-4 (1.2e-6) for all optimizers in the NMT (LM) task. For adaptive optimizers, we set $(\beta_1, \beta_2)$ to (0.9, 0.98) in the NMT task and (0.9, 0.999) in the LM task. For the LM task, the general dropout rate is set to 0.4.

**CV**

- SGD/Adam/AdamW: We adopt the same experimental setup in AdaHessian [34]. For SGD, the initial learning rate is 0.1 and the momentum is set to 0.9. For Adam, the initial learning rate is set to 0.001 and the epsilon is set to 1e-8. For AdamW, the initial learning rate is set to 0.005 and the epsilon is set to 1e-8.

- AdaBelief: We explore the best learning rate for ResNet20/32 on Cifar10 and ResNet18 on ImageNet, respectively. Finally, the initial learning rate is set to be 0.01 for ResNet20 on Cifar10 and 0.005 for ResNet32/ResNet18 on Cifar10/ImageNet. The epsilon is set to 1e-8.

- AdaHessian: We use the recommended configuration as much as possible from AdaHessian [34]. The Hessian power is set to 1. The initial learning rate is 0.15 when training on both Cifar10 and ImageNet, as recommended in Yao et al. [34]. The epsilon is set to 1e-4.

- AGD: We conduct a grid search of $\delta$ and the learning rate. The choice of $\delta$ is among {1e-8, 1e-7, 1e-6, 1e-5, 1e-4, 1e-3, 1e-2} and the search range for the learning rate is from 1e-4 to 1e-2. Finally, we choose the learning rate to 0.007 and $\delta$ to 1e-2 for Cifar10 task, and learning rate to 0.0004 and $\delta$ to 1e-5 for ImageNet task.

The weight decay for all optimizers is set to 0.0005 on Cifar10 and 0.0001 on ImageNet. $\beta_1 = 0.9$ and $\beta_2 = 0.999$ are for all adaptive optimizers.

---

[4]https://github.com/juntang-zhuang/Adabelief-Optimizer
[5]https://github.com/amirgholami/adahessian

**RecSys**    Note that we implement the optimizers for training on our internal distributed environment.

- SGD: We search for the learning rate among {1e-4, 1e-3, 1e-2, 0.1, 1} and choose the best results (0.1 for the Avazu task and 1e-3 for the Criteo task).
- Adam/AdaBelief: We search the learning rate among {1e-5, 1e-4, 1e-3, 1e-2} and the epsilon among {1e-16, 1e-14, 1e-12, 1e-10, 1e-8, 1e-6}. For the Avazu task, the best learning rate/epsilon is 1e-4/1e-8 for Adam and 1e-4/1e-16 for AdaBelief. For the Criteo task, the best learning rate/epsilon is 1e-3/1e-8 for Adam and 1e-3/1e-16 for AdaBelief.
- AdaHessian: We search the learning rate among {1e-5, 1e-4, 1e-3, 1e-2} and the epsilon among {1e-16, 1e-14, 1e-12, 1e-10, 1e-8, 1e-6}. The best learning rate/epsilon is 1e-4/1e-8 for the Avazu task and 1e-3/1e-6 for the Criteo task. The block size and the Hessian power are set to 1.
- AGD: We search the learning rate among {1e-5, 1e-4, 1e-3} and $\delta$ among {1e-12, 1e-10, 1e-8, 1e-6, 1e-4, 1e-2}. The best learning rate/$\delta$ is 1e-4/1e-4 for the Avazu task and 1e-4/1e-8 for the Criteo task.

$\beta_1 = 0.9$ and $\beta_2 = 0.999$ are for all adaptive optimizers.

## A.2    AGD vs. AdaBound

Table 7: The performance of AGD and AdaBound across different tasks.

| Optimizer | 3-Layer LSTM, test PPL (lower is better) | ResNet18 on ImageNet, Top-1 accuracy |
|---|---|---|
| AdaBound | 63.60 [39] | 68.13 (100 epochs) [8] |
| AGD | **60.89** (better) | **70.19** (90 epochs, still better) |

## A.3    Numerical Experiments

In our numerical experiments, we employ the same learning rate across optimizers. While a larger learning rate could accelerate convergence, as shown in Figures 7a and 7b, we also note that it could lead to unstable training. To investigate the largest learning rate that each optimizer could handle for the Beale function, we perform a search across the range of {1e-5, 1e-4, ..., 1, 10}. The optimization trajectories are displayed in Figure 7c, and AGD demonstrates slightly superior performance. Nonetheless, we argue that the learning rate selection outlined in Section 3.3 presents a more appropriate representation.

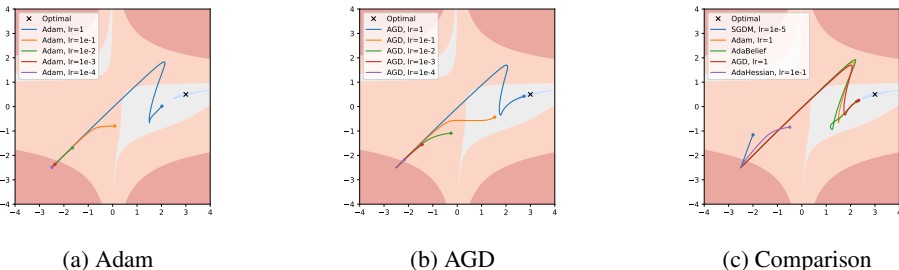

|  (a) Adam | (b) AGD | (c) Comparison |

Figure 7: Optimization trajectories on Beale function using various learning rates.

## A.4    AGD with AMSGrad condition

Algorithm 2 summarizes the AGD optimizer with AMSGrad condition. The AMSGrad condition is usually used to ensure the convergence. We empirically show that adding AMSGrad condition to AGD will slightly degenerate AGD's performance, as listed in Table 8. For AGD with AMSGrad condition, we search for the best learning rate among {0.001, 0.003, 0.005, 0.007, 0.01} and best $\delta$ within {1e-2, 1e-4, 1e-6, 1e-8}. We find the best learning rate is 0.007 and best $\delta$ is 1e-2, which are the same as AGD without AMSGrad condition.

## A.5    AGD for ResNet18 on Cifar10

Since the SOTA accuracy [6] for ResNet18 on Cifar10 is 95.55% using SGD optimizer with a cosine learning rate schedule, we also test the performance of AGD for ResNet18 on Cifar10. We find AGD can achieve 95.79%

---

[6]https://paperswithcode.com/sota/stochastic-optimization-on-cifar-10-resnet-18

**Algorithm 2** AGD with AMSGrad condition

---

1: **Input:** parameters $\beta_1, \beta_2, \delta, \boldsymbol{w}_1 \in \mathbb{R}^n$, step size $\alpha_t$, initialize $\boldsymbol{m}_0 = \boldsymbol{0}, \boldsymbol{b}_0 = \boldsymbol{0}$
2: **for** $t = 1$ **to** $T$ **do**
3:     $\boldsymbol{g}_t = \nabla f_t(\boldsymbol{w}_t)$
4:     $\boldsymbol{m}_t \leftarrow \beta_1 \boldsymbol{m}_{t-1} + (1 - \beta_1)\boldsymbol{g}_t$
5:     $\boldsymbol{s}_t = \begin{cases} \boldsymbol{m}_1/(1 - \beta_1) & t = 1 \\ \boldsymbol{m}_t/(1 - \beta_1^t) - \boldsymbol{m}_{t-1}/(1 - \beta_1^{t-1}) & t > 1 \end{cases}$
6:     $\boldsymbol{b}_t \leftarrow \beta_2 \boldsymbol{b}_{t-1} + (1 - \beta_2)\boldsymbol{s}_t^2$
7:     $\boldsymbol{b}_t \leftarrow \max(\boldsymbol{b}_t, \boldsymbol{b}_{t-1})$         // AMSGrad condtion
8:     $\boldsymbol{w}_{t+1} = \boldsymbol{w}_t - \alpha_t \dfrac{\sqrt{1 - \beta_2^t}}{1 - \beta_1^t} \dfrac{\boldsymbol{m}_t}{\max(\sqrt{\boldsymbol{b}_t}, \delta\sqrt{1 - \beta_2^t})}$
9: **end for**

---

Table 8: Top-1 accuracy for AGD with and without AMSGrad condition when trained with ResNet20 on Cifar10.

| Optimizer | AGD | AGD + AMSGrad |
|---|---|---|
| Accuracy | $\mathbf{92.35 \pm .24}$ | $92.25 \pm 0.11$ |

accuracy at lr = 0.001 and $\delta$ = 1e-2 when using the same training configuration as the experiment of SGD in Moreau et al. [24], which exceeds the current SOTA result.

### A.6 Robustness to hyperparameters

We test the performance of AGD and Adam with respect to $\delta$ (or $\epsilon$) and learning rate. The experiments are performed with ResNet20 on Cifar10 and the results are shown in Figure 8. Compared to Adam, AGD shows better robustness to the change of hyperparameters.

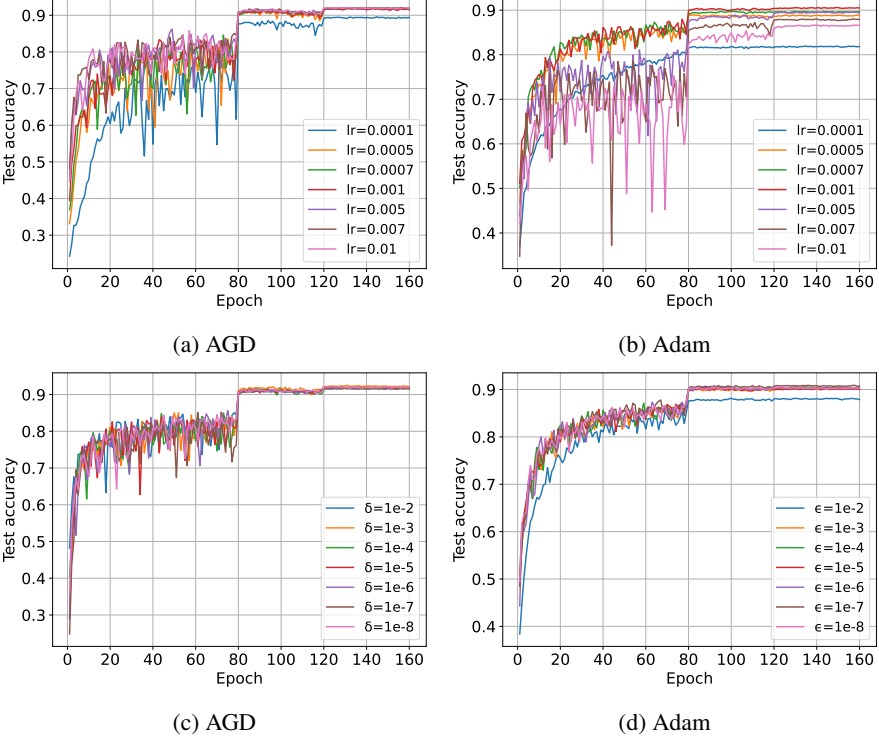

Figure 8: Test accuracy of ResNet20 on Cifar10, trained with AGD and Adam using different $\delta$ (or $\epsilon$) and learning rate. For (a) and (b), we choose learning rate as 0.007 and 0.001, respectively. For (c) and (d), we set $\delta$ (or $\epsilon$) to be 1e-2 and 1e-8, respectively.

# B  Proof of Theorem 1

*Proof.* Denote $\hat{\alpha}_t = \alpha_t \frac{\sqrt{1-\beta_2^t}}{1-\beta_{1,t}^t}$ and $\boldsymbol{v}_t = \max(\sqrt{\boldsymbol{b}_t}, \delta\sqrt{1-\beta_2^t})$, we have the following lemmas.

**Lemma 1.** *For the parameter settings and assumptions in Theorem 1 or Theorem 2, we have*

$$\hat{\alpha}_t > \hat{\alpha}_{t+1}, \ t \in [T],$$

*where $\hat{\alpha}_t = \alpha_t \frac{\sqrt{1-\beta_2^t}}{1-\beta_{1,t}^t}$.*

*Proof.* Since $\beta_{1,t}$ is non-increasing with respect to $t$, we have $\beta_{1,t+1}^{t+1} \leq \beta_{1,t+1}^t \leq \beta_{1,t}^t$. Hence, $\beta_{1,t}^t$ is non-increasing and $\frac{1}{1-\beta_{1,t}^t}$ is non-increasing. Thus, we only need to prove $\phi(t) = \alpha_t\sqrt{1-\beta_2^t}$ is decreasing. Since the proof is trivial when $\beta_2 = 0$, we only need to consider the case where $\beta_2 \in (0,1)$. Taking the derivative of $\phi(t)$, we have

$$\phi'(t) = -\frac{\alpha}{2}t^{-\frac{3}{2}}(1-\beta_2^t)^{\frac{1}{2}} - \frac{\alpha}{2}t^{-\frac{1}{2}}(1-\beta_2^t)^{-\frac{1}{2}}\beta_2^t\log_e\beta_2 = -\frac{\alpha}{2}t^{-\frac{3}{2}}(1-\beta_2^t)^{-\frac{1}{2}}\underbrace{(1-\beta_2^t + t\beta_2^t\log_e\beta_2)}_{\psi(t)}.$$

Since

$$\psi'(t) = -\beta_2^t\log_e\beta_2 + \beta_2^t\log_e\beta_2 + t\beta_2^t(\log_e\beta_2)^2 = t\beta_2^t(\log_e\beta_2)^2,$$

we have $\psi'(t) > 0$ when $t > 0$ and $\psi'(t) = 0$ when $t = 0$. Combining $\psi(0) = 0$, we get $\psi(t) > 0$ when $t > 0$. Thus, we have $\phi'(t) < 0$, which completes the proof. $\square$

**Lemma 2.** *For the parameter settings and assumptions in Theorem 1 or Theorem 2, we have*

$$\|\sqrt{\boldsymbol{v}_t}\|^2 < \frac{n(2G_\infty + \delta)}{(1-\beta_1)^2}, \ t \in [T],$$

*where $\boldsymbol{v}_t = \max(\sqrt{\boldsymbol{b}_t}, \delta\sqrt{1-\beta_2^t})$.*

*Proof.*

$$\|\boldsymbol{m}_t\|_\infty = \|\sum_{i=1}^{t}(1-\beta_{1,t-i+1})\boldsymbol{g}_{t-i+1}\prod_{j=1}^{i-1}\beta_{1,t-j+1}\|_\infty \leq \|\sum_{i=1}^{t}\boldsymbol{g}_{t-i+1}\beta_1^{i-1}\|_\infty \leq \frac{G_\infty}{1-\beta_1},$$

$$\|\boldsymbol{s}_t\|_\infty \leq \begin{cases} \frac{\|\boldsymbol{m}_1\|_\infty}{1-\beta_{1,t}} \leq \frac{G_\infty}{(1-\beta_1)^2} < \frac{2G_\infty}{(1-\beta_1)^2} & t = 1, \\ \frac{\|\boldsymbol{m}_t\|_\infty}{1-\beta_{1,t}^t} + \frac{\|\boldsymbol{m}_{t-1}\|_\infty}{1-\beta_{1,t}^{t-1}} \leq \frac{2G_\infty}{(1-\beta_1)^2} & t > 1, \end{cases}$$

$$\|\boldsymbol{b}_t\|_\infty = \|(1-\beta_2)\sum_{i=1}^{t}\boldsymbol{s}_{t-i+1}^2\beta_2^{i-1}\|_\infty \leq \frac{4G_\infty^2}{(1-\beta_1)^4},$$

$$\|\sqrt{\boldsymbol{v}_t}\|^2 = \sum_{i=1}^{n}v_{t,i} < n(\|\sqrt{\boldsymbol{b}_t}\|_\infty + \delta) \leq \frac{n(2G_\infty + \delta)}{(1-\beta_1)^2}.$$

$\square$

By assumptions 2, 4, Lemma 1 and Lemma 2, $\forall t \in [T]$, we have

$$\|\boldsymbol{m}_t\|_\infty \leq \frac{G_\infty}{1-\beta_1}, \quad \|\boldsymbol{v}_t\|_\infty \leq \frac{2G_\infty}{(1-\beta_1)^2}, \quad \frac{\hat{\alpha}_t}{v_{t,i}} > \frac{\hat{\alpha}_{t+1}}{v_{t+1,i}}, \ \forall i \in [n]. \tag{6}$$

Following Yang et al. [33], Chen et al. [9], Zhou et al. [37], we define an auxiliary sequence $\{\boldsymbol{u}_t\}$: $\forall t \geq 2$,

$$\boldsymbol{u}_t = \boldsymbol{w}_t + \frac{\beta_{1,t}}{1-\beta_{1,t}}(\boldsymbol{w}_t - \boldsymbol{w}_{t-1}) = \frac{1}{1-\beta_{1,t}}\boldsymbol{w}_t - \frac{\beta_{1,t}}{1-\beta_{1,t}}\boldsymbol{w}_{t-1}. \tag{7}$$

Hence, we have

$$\boldsymbol{u}_{t+1} - \boldsymbol{u}_t = \left( \frac{1}{1-\beta_{1,t+1}} - \frac{1}{1-\beta_{1,t}} \right) \boldsymbol{w}_{t+1} - \left( \frac{\beta_{1,t+1}}{1-\beta_{1,t+1}} - \frac{\beta_{1,t}}{1-\beta_{1,t}} \right) \boldsymbol{w}_t$$

$$+ \frac{1}{1-\beta_{1,t}}(\boldsymbol{w}_{t+1} - \boldsymbol{w}_t) - \frac{\beta_{1,t}}{1-\beta_{1,t}}(\boldsymbol{w}_t - \boldsymbol{w}_{t-1})$$

$$= \left( \frac{1}{1-\beta_{1,t+1}} - \frac{1}{1-\beta_{1,t}} \right) \left( \boldsymbol{w}_t - \hat{\alpha}_t \frac{\boldsymbol{m}_t}{\boldsymbol{v}_t} \right) - \left( \frac{\beta_{1,t+1}}{1-\beta_{1,t+1}} - \frac{\beta_{1,t}}{1-\beta_{1,t}} \right) \boldsymbol{w}_t$$

$$- \frac{\hat{\alpha}_t}{1-\beta_{1,t}} \frac{\boldsymbol{m}_t}{\boldsymbol{v}_t} + \frac{\beta_{1,t}\hat{\alpha}_{t-1}}{1-\beta_{1,t}} \frac{\boldsymbol{m}_{t-1}}{\boldsymbol{v}_{t-1}}$$

$$= \left( \frac{1}{1-\beta_{1,t}} - \frac{1}{1-\beta_{1,t+1}} \right) \hat{\alpha}_t \frac{\boldsymbol{m}_t}{\boldsymbol{v}_t} - \frac{\hat{\alpha}_t}{1-\beta_{1,t}} \left( \beta_{1,t}\frac{\boldsymbol{m}_{t-1}}{\boldsymbol{v}_t} + (1-\beta_{1,t})\frac{\boldsymbol{g}_t}{\boldsymbol{v}_t} \right) + \frac{\beta_{1,t}\hat{\alpha}_{t-1}}{1-\beta_{1,t}} \frac{\boldsymbol{m}_{t-1}}{\boldsymbol{v}_{t-1}}$$

$$= \left( \frac{1}{1-\beta_{1,t}} - \frac{1}{1-\beta_{1,t+1}} \right) \hat{\alpha}_t \frac{\boldsymbol{m}_t}{\boldsymbol{v}_t} + \frac{\beta_{1,t}}{1-\beta_{1,t}} \left( \frac{\hat{\alpha}_{t-1}}{\boldsymbol{v}_{t-1}} - \frac{\hat{\alpha}_t}{\boldsymbol{v}_t} \right) \boldsymbol{m}_{t-1} - \hat{\alpha}_t \frac{\boldsymbol{g}_t}{\boldsymbol{v}_t}.$$

$$(8)$$

By assumption 1 and Equation (8), we have

$$f(\boldsymbol{u}_{t+1}) \le f(\boldsymbol{u}_t) + \langle \nabla f(\boldsymbol{u}_t), \boldsymbol{u}_{t+1} - \boldsymbol{u}_t \rangle + \frac{L}{2}\|\boldsymbol{u}_{t+1} - \boldsymbol{u}_t\|^2$$

$$= f(\boldsymbol{u}_t) + \langle \nabla f(\boldsymbol{w}_t), \boldsymbol{u}_{t+1} - \boldsymbol{u}_t \rangle + \langle \nabla f(\boldsymbol{u}_t) - \nabla f(\boldsymbol{w}_t), \boldsymbol{u}_{t+1} - \boldsymbol{u}_t \rangle + \frac{L}{2}\|\boldsymbol{u}_{t+1} - \boldsymbol{u}_t\|^2$$

$$= f(\boldsymbol{u}_t) + \left\langle \nabla f(\boldsymbol{w}_t), \left( \frac{1}{1-\beta_{1,t}} - \frac{1}{1-\beta_{1,t+1}} \right) \hat{\alpha}_t \frac{\boldsymbol{m}_t}{\boldsymbol{v}_t} \right\rangle + \frac{\beta_{1,t}}{1-\beta_{1,t}} \left\langle \nabla f(\boldsymbol{w}_t), \left( \frac{\hat{\alpha}_{t-1}}{\boldsymbol{v}_{t-1}} - \frac{\hat{\alpha}_t}{\boldsymbol{v}_t} \right) \boldsymbol{m}_{t-1} \right\rangle$$

$$- \hat{\alpha}_t \left\langle \nabla f(\boldsymbol{w}_t), \frac{\boldsymbol{g}_t}{\boldsymbol{v}_t} \right\rangle + \langle \nabla f(\boldsymbol{u}_t) - \nabla f(\boldsymbol{w}_t), \boldsymbol{u}_{t+1} - \boldsymbol{u}_t \rangle + \frac{L}{2}\|\boldsymbol{u}_{t+1} - \boldsymbol{u}_t\|^2.$$

$$(9)$$

Rearranging Equation (9) and taking expectation both sides, by assumption 3 and Equation (6), we get

$$\frac{(1-\beta_1)^2\hat{\alpha}_t}{2G_\infty} \mathbf{E}[\|\nabla f(\boldsymbol{w}_t)\|^2] \le \hat{\alpha}_t \mathbf{E}\left[ \left\langle \nabla f(\boldsymbol{w}_t), \frac{\nabla f(\boldsymbol{w}_t)}{\boldsymbol{v}_t} \right\rangle \right]$$

$$\le \mathbf{E}[f(\boldsymbol{u}_t) - f(\boldsymbol{u}_{t+1})] + \underbrace{\mathbf{E}\left[ \left\langle \nabla f(\boldsymbol{w}_t), \left( \frac{1}{1-\beta_{1,t}} - \frac{1}{1-\beta_{1,t+1}} \right) \hat{\alpha}_t \frac{\boldsymbol{m}_t}{\boldsymbol{v}_t} \right\rangle \right]}_{P_1}$$

$$+ \frac{\beta_{1,t}}{1-\beta_{1,t}} \underbrace{\mathbf{E}\left[ \left\langle \nabla f(\boldsymbol{w}_t), \left( \frac{\hat{\alpha}_{t-1}}{\boldsymbol{v}_{t-1}} - \frac{\hat{\alpha}_t}{\boldsymbol{v}_t} \right) \boldsymbol{m}_{t-1} \right\rangle \right]}_{P_2}$$

$$+ \underbrace{\mathbf{E}\left[ \langle \nabla f(\boldsymbol{u}_t) - \nabla f(\boldsymbol{w}_t), \boldsymbol{u}_{t+1} - \boldsymbol{u}_t \rangle \right]}_{P_3} + \frac{L}{2} \underbrace{\mathbf{E}\left[ \|\boldsymbol{u}_{t+1} - \boldsymbol{u}_t\|^2 \right]}_{P_4}.$$

$$(10)$$

To further bound Equation (10), we need the following lemma.

**Lemma 3.** *For the sequence $\{\boldsymbol{u}_t\}$ defined as Equation (7), $\forall t \ge 2$, we have*

$$\|\boldsymbol{u}_{t+1} - \boldsymbol{u}_t\| \le \frac{\sqrt{n}G_\infty}{\delta} \left( \frac{\hat{\alpha}_t\beta_{1,t}}{(1-\beta_1)^3} + \frac{\hat{\alpha}_{t-1}\beta_{1,t}}{(1-\beta_1)^2} + \hat{\alpha}_t \right),$$

$$\|\boldsymbol{u}_{t+1} - \boldsymbol{u}_t\|^2 \le \frac{3nG_\infty^2}{\delta^2} \left( \frac{\hat{\alpha}_t^2\beta_{1,t}^2}{(1-\beta_1)^6} + \frac{\hat{\alpha}_{t-1}^2\beta_{1,t}^2}{(1-\beta_1)^4} + \hat{\alpha}_t^2 \right).$$

*Proof.* Since $\forall t \in [T], \forall i \in [n], 1/(1-\beta_{1,t}) \ge 1/(1-\beta_{1,t+1}), \boldsymbol{v}_t \ge \delta\sqrt{1-\beta_2}, \hat{\alpha}_{t-1}/v_{t-1,i} > \hat{\alpha}_t/v_{t,i}.$
By Equation (8), we have

$$\|\boldsymbol{u}_{t+1} - \boldsymbol{u}_t\| \le \hat{\alpha}_t \frac{\sqrt{n}G_\infty}{(1-\beta_1)\delta\sqrt{1-\beta_2}} \left( \frac{\beta_{1,t} - \beta_{1,t+1}}{(1-\beta_{1,t})(1-\beta_{1,t+1})} \right) + \frac{\beta_{1,t}}{1-\beta_{1,t}}\hat{\alpha}_{t-1} \frac{\sqrt{n}G_\infty}{(1-\beta_1)\delta\sqrt{1-\beta_2}} + \hat{\alpha}_t \frac{\sqrt{n}G_\infty}{\delta\sqrt{1-\beta_2}}$$

$$\le \frac{\sqrt{n}G_\infty}{\delta\sqrt{1-\beta_2}} \left( \frac{\hat{\alpha}_t\beta_{1,t}}{(1-\beta_1)^3} + \frac{\hat{\alpha}_{t-1}\beta_{1,t}}{(1-\beta_1)^2} + \hat{\alpha}_t \right),$$

$$\|\boldsymbol{u}_{t+1} - \boldsymbol{u}_t\|^2 \le \frac{3nG_\infty^2}{\delta^2(1-\beta_2)} \left( \frac{\hat{\alpha}_t^2\beta_{1,t}^2}{(1-\beta_1)^6} + \frac{\hat{\alpha}_{t-1}^2\beta_{1,t}^2}{(1-\beta_1)^4} + \hat{\alpha}_t^2 \right),$$

where the last inequality follows from Cauchy-Schwartz inequality. This completes the proof. $\qquad\square$

Now we bound $P_1$, $P_2$, $P_3$ and $P_4$ of Equation (10), separately. By assumptions 1, 2, Equation (6) and Lemma 3, we have

$$P_1 \leq \hat{\alpha}_t \left( \frac{1}{1 - \beta_{1,t}} - \frac{1}{1 - \beta_{1,t+1}} \right) \mathbf{E}\left[ \|\nabla f(\boldsymbol{w}_t)\| \| \frac{\boldsymbol{m}_t}{\boldsymbol{v}_t} \| \right]$$

$$\leq \hat{\alpha}_t \left( \frac{\beta_{1,t} - \beta_{1,t+1}}{(1 - \beta_{1,t})(1 - \beta_{1,t+1})} \right) \frac{nG_\infty^2}{\delta\sqrt{1 - \beta_2}(1 - \beta_1)} \leq \frac{nG_\infty^2}{(1 - \beta_1)^3 \delta\sqrt{1 - \beta_2}} \hat{\alpha}_t(\beta_{1,t} - \beta_{1,t+1}),$$

$$P_2 = \mathbf{E}\left[ \sum_{i=1}^n \nabla_i f(\boldsymbol{w}_t) m_{t-1,i} (\frac{\hat{\alpha}_{t-1}}{v_{t-1,i}} - \frac{\hat{\alpha}_t}{v_{t,i}}) \right] \leq \frac{G_\infty^2}{1 - \beta_1} \sum_{i=1}^n (\frac{\hat{\alpha}_{t-1}}{v_{t-1,i}} - \frac{\hat{\alpha}_t}{v_{t,i}}),$$

$$P_3 \leq \mathbf{E}\left[ \|\nabla f(\boldsymbol{u}_t) - \nabla f(\boldsymbol{w}_t)\| \|\boldsymbol{u}_{t+1} - \boldsymbol{u}_t\| \right] \leq L\mathbf{E}\left[ \|\boldsymbol{u}_t - \boldsymbol{w}_t\| \|\boldsymbol{u}_{t+1} - \boldsymbol{u}_t\| \right]$$

$$= L\hat{\alpha}_{t-1} \frac{\beta_{1,t}}{1 - \beta_{1,t}} \mathbf{E}\left[ \|\frac{\boldsymbol{m}_{t-1}}{\boldsymbol{v}_{t-1}}\| \|\boldsymbol{u}_{t+1} - \boldsymbol{u}_t\| \right] \leq \frac{LnG_\infty^2}{(1 - \beta_1)^2 \delta^2 (1 - \beta_2)} \left( \frac{\hat{\alpha}_{t-1}\hat{\alpha}_t\beta_{1,t}^2}{(1 - \beta_1)^3} + \frac{\hat{\alpha}_{t-1}^2\beta_{1,t}^2}{(1 - \beta_1)^2} + \hat{\alpha}_{t-1}\hat{\alpha}_t\beta_{1,t} \right)$$

$$< \frac{LnG_\infty^2}{(1 - \beta_1)^2 \delta^2 (1 - \beta_2)} \left( \frac{\hat{\alpha}_{t-1}^2}{(1 - \beta_1)^3} + \frac{\hat{\alpha}_{t-1}^2}{(1 - \beta_1)^2} + \hat{\alpha}_{t-1}^2 \right) < \frac{3LnG_\infty^2}{(1 - \beta_1)^5 \delta^2 (1 - \beta_2)} \hat{\alpha}_{t-1}^2,$$

$$P_4 \leq \frac{3nG_\infty^2}{\delta^2 (1 - \beta_2)} \left( \frac{\hat{\alpha}_t^2\beta_{1,t}^2}{(1 - \beta_1)^6} + \frac{\hat{\alpha}_{t-1}^2\beta_{1,t}^2}{(1 - \beta_1)^4} + \hat{\alpha}_t^2 \right) < \frac{3nG_\infty^2}{\delta^2 (1 - \beta_2)} \left( \frac{\hat{\alpha}_t^2}{(1 - \beta_1)^6} + \frac{\hat{\alpha}_{t-1}^2}{(1 - \beta_1)^4} + \hat{\alpha}_t^2 \right)$$

$$< \frac{9nG_\infty^2}{(1 - \beta_1)^6 \delta^2 (1 - \beta_2)} \hat{\alpha}_{t-1}^2.$$

(11)

Replacing $P_1$, $P_2$, $P_3$ and $P_4$ of Equation (10) with Equation (11) and telescoping Equation (10) for $t = 2$ to $T$, we have

$$\sum_{t=2}^T \frac{(1 - \beta_1)^2 \hat{\alpha}_t}{2G_\infty} \mathbf{E}\left[ \|\nabla f(\boldsymbol{w}_t)\|^2 \right] < \mathbf{E}\left[ f(\boldsymbol{u}_2) - f(\boldsymbol{u}_{T+1}) \right] + \frac{nG_\infty^2}{(1 - \beta_1)^3 \delta\sqrt{1 - \beta_2}} \sum_{t=2}^T \hat{\alpha}_t(\beta_{1,t} - \beta_{1,t+1})$$

$$+ \frac{\beta_1 G_\infty^2}{(1 - \beta_1)^2} \sum_{i=1}^n \left( \frac{\hat{\alpha}_1}{v_{1,i}} - \frac{\hat{\alpha}_T}{v_{T,i}} \right) + \frac{3LnG_\infty^2}{(1 - \beta_1)^5 \delta^2 (1 - \beta_2)} \sum_{t=2}^T \hat{\alpha}_{t-1}^2 + \frac{9LnG_\infty^2}{2(1 - \beta_1)^6 \delta^2 (1 - \beta_2)} \sum_{t=2}^T \hat{\alpha}_{t-1}^2$$

$$< \mathbf{E}\left[ f(\boldsymbol{u}_2) \right] - f(\boldsymbol{w}^*) + \frac{nG_\infty^2}{(1 - \beta_1)^3 \delta\sqrt{1 - \beta_2}} \sum_{t=1}^T \hat{\alpha}_t(\beta_{1,t} - \beta_{1,t+1}) + \frac{\alpha\beta_1 nG_\infty^2}{(1 - \beta_1)^3 \delta}$$

$$+ \frac{15LnG_\infty^2}{2(1 - \beta_1)^6 \delta^2 (1 - \beta_2)} \sum_{t=1}^T \hat{\alpha}_t^2.$$

(12)

Since

$$\sum_{t=2}^T \hat{\alpha}_t = \sum_{t=2}^T \frac{\alpha}{\sqrt{t}} \frac{\sqrt{1 - \beta_2^t}}{1 - \beta_{1,t}^t} \geq \alpha\sqrt{1 - \beta_2} \sum_{t=2}^T \frac{1}{\sqrt{t}}$$

$$= \alpha\sqrt{1 - \beta_2} \left( \int_2^3 \frac{1}{\sqrt{2}} ds + \cdots + \int_{T-1}^T \frac{1}{\sqrt{T}} ds \right) > \alpha\sqrt{1 - \beta_2} \int_2^T \frac{1}{\sqrt{s}} ds$$

$$= 2\alpha\sqrt{1 - \beta_2} \left( \sqrt{T} - \sqrt{2} \right),$$

$$\sum_{t=1}^T \hat{\alpha}_t^2 = \sum_{t=1}^T \frac{\alpha^2}{t} \frac{1 - \beta_2^t}{(1 - \beta_{1,t})^2} \leq \frac{\alpha^2}{(1 - \beta_1)^2} \sum_{t=1}^T \frac{1}{t}$$

$$= \frac{\alpha^2}{(1 - \beta_1)^2} \left( 1 + \int_2^3 \frac{1}{2} ds + \cdots + \int_{T-1}^T \frac{1}{T} ds \right) < \frac{\alpha^2}{(1 - \beta_1)^2} \left( 1 + \int_2^T \frac{1}{s - 1} ds \right)$$

$$= \frac{\alpha^2}{(1 - \beta_1)^2} \left( \log(T - 1) + 1 \right) < \frac{\alpha^2}{(1 - \beta_1)^2} \left( \log T + 1 \right),$$

(13)

$$\mathbf{E}\left[ f(\boldsymbol{u}_2) \right] \leq f(\boldsymbol{w}_1) + \mathbf{E}\left[ \langle \nabla f(\boldsymbol{w}_1), \boldsymbol{u}_2 - \boldsymbol{w}_1 \rangle \right] + \frac{L}{2} \mathbf{E}\left[ \|\boldsymbol{u}_2 - \boldsymbol{w}_1\|^2 \right]$$

$$= f(\boldsymbol{w}_1) - \frac{\hat{\alpha}_1}{1 - \beta_{1,2}} \mathbf{E}\left[ \langle \nabla f(\boldsymbol{w}_1), \frac{\boldsymbol{m}_1}{\boldsymbol{v}_1} \rangle \right] + \frac{L\hat{\alpha}_1^2}{2(1 - \beta_{1,2})^2} \mathbf{E}\left[ \|\frac{\boldsymbol{m}_1}{\boldsymbol{v}_1}\|^2 \right]$$

$$\leq f(\boldsymbol{w}_1) + \frac{\alpha\sqrt{1 - \beta_2}}{(1 - \beta_1)^2} \mathbf{E}\left[ \|\nabla f(\boldsymbol{w}_1)\| \| \frac{\boldsymbol{m}_1}{\boldsymbol{v}_1} \| \right] + \frac{L\alpha^2(1 - \beta_2)}{2(1 - \beta_1)^4} \mathbf{E}\left[ \|\frac{\boldsymbol{m}_1}{\boldsymbol{v}_1}\|^2 \right]$$

$$\leq f(\boldsymbol{w}_1) + \frac{\alpha nG_\infty^2}{(1 - \beta_1)^2 \delta} + \frac{L\alpha^2 nG_\infty^2}{2(1 - \beta_1)^4 \delta^2} \leq f(\boldsymbol{w}_1) + \frac{nG_\infty^2 \alpha}{2(1 - \beta_1)^4 \delta^2} (2\delta + L\alpha),$$

substituting Equation (13) into Equation (12), we finish the proof. $\qquad\square$

## C  Proof of Corollary 1

*Proof.* Since $\beta_{1,t} = \beta_1/\sqrt{t}$, we have

$$\sum_{t=1}^{T} \hat{\alpha}_t(\beta_{1,t} - \beta_{1,t+1}) \leq \sum_{t=1}^{T} \hat{\alpha}_t \beta_{1,t} = \sum_{t=1}^{T} \frac{\alpha}{\sqrt{t}} \frac{\sqrt{1-\beta_2^t}}{1-\beta_{1,t}^t} \beta_{1,t} < \frac{\alpha}{1-\beta_1} \sum_{t=1}^{T} \frac{1}{t} < \frac{\alpha}{1-\beta_1}(\log T + 1).$$

(14)

Substituting Equation (14) into Equation (5), we have

$$\min_{t \in [T]} \mathbf{E}\left[\|\nabla f(\boldsymbol{w}_t)\|^2\right] < \frac{G_\infty}{\alpha(1-\beta_1)^2(1-\beta_2)^2}\left(f(\boldsymbol{w}_1) - f(\boldsymbol{w}^*) + \frac{nG_\infty^2\alpha}{(1-\beta_1)^8\delta^2}(2\delta + 8L\alpha)\right.$$

$$\left. + \frac{\alpha\beta_1 nG_\infty^2}{(1-\beta_1)^3\delta}\right)\frac{1}{\sqrt{T}-\sqrt{2}} + \frac{nG_\infty^3}{(1-\beta_2)^2(1-\beta_1)^{10}\delta^2}\left(\frac{15}{2}L\alpha + \delta\right)\frac{\log T}{\sqrt{T}-\sqrt{2}}.$$

This completes the proof. $\qquad\square$

## D  Proof of Theorem 2

*Proof.* Denote $\hat{\alpha}_t = \alpha_t \frac{\sqrt{1-\beta_2^t}}{1-\beta_{1,t}^t}$ and $\boldsymbol{v}_t = \max(\sqrt{\boldsymbol{b}_t}, \delta\sqrt{1-\beta_2^t})$, then

$$\|\sqrt{\boldsymbol{v}_t}(\boldsymbol{w}_{t+1} - \boldsymbol{w}^*)\|^2 = \|\sqrt{\boldsymbol{v}_t}(\boldsymbol{w}_t - \hat{\alpha}_t \frac{\boldsymbol{m}_t}{\boldsymbol{v}_t} - \boldsymbol{w}^*)\|^2 = \|\sqrt{\boldsymbol{v}_t}(\boldsymbol{w}_t - \boldsymbol{w}^*)\|^2 - 2\hat{\alpha}_t \langle \boldsymbol{w}_t - \boldsymbol{w}^*, \boldsymbol{m}_t \rangle + \hat{\alpha}_t^2 \|\frac{\boldsymbol{m}_t}{\sqrt{\boldsymbol{v}_t}}\|^2$$

$$= \|\sqrt{\boldsymbol{v}_t}(\boldsymbol{w}_t - \boldsymbol{w}^*)\|^2 + \hat{\alpha}_t^2 \|\frac{\boldsymbol{m}_t}{\sqrt{\boldsymbol{v}_t}}\|^2 - 2\hat{\alpha}_t\beta_{1,t} \langle \boldsymbol{w}_t - \boldsymbol{w}^*, \boldsymbol{m}_{t-1} \rangle - 2\hat{\alpha}_t(1-\beta_{1,t}) \langle \boldsymbol{w}_t - \boldsymbol{w}^*, \boldsymbol{g}_t \rangle.$$

(15)

Rearranging Equation (15), we have

$$\langle \boldsymbol{w}_t - \boldsymbol{w}^*, \boldsymbol{g}_t \rangle = \frac{1}{1-\beta_{1,t}}\left[\frac{1}{2\hat{\alpha}_t}(\|\sqrt{\boldsymbol{v}_t}(\boldsymbol{w}_t - \boldsymbol{w}^*)\|^2 - \|\sqrt{\boldsymbol{v}_t}(\boldsymbol{w}_{t+1} - \boldsymbol{w}^*)\|^2) - \beta_{1,t} \langle \boldsymbol{w}_t - \boldsymbol{w}^*, \boldsymbol{m}_{t-1} \rangle + \frac{\hat{\alpha}_t}{2}\|\frac{\boldsymbol{m}_t}{\sqrt{\boldsymbol{v}_t}}\|^2\right]$$

$$\leq \frac{1}{1-\beta_1}\left[\frac{1}{2\hat{\alpha}_t}(\|\sqrt{\boldsymbol{v}_t}(\boldsymbol{w}_t - \boldsymbol{w}^*)\|^2 - \|\sqrt{\boldsymbol{v}_t}(\boldsymbol{w}_{t+1} - \boldsymbol{w}^*)\|^2) + \frac{\beta_{1,t}}{2\hat{\alpha}_t}\|\boldsymbol{w}_t - \boldsymbol{w}^*\|^2\right.$$

$$\left. + \frac{\beta_{1,t}\hat{\alpha}_t}{2}\|\boldsymbol{m}_{t-1}\|^2 + \frac{\hat{\alpha}_t}{2}\|\frac{\boldsymbol{m}_t}{\sqrt{\boldsymbol{v}_t}}\|^2\right],$$

where the first inequality follows from Cauchy-Schwartz inequality and $ab \leq \frac{1}{2}(a^2 + b^2)$. Hence, the regret

$$\sum_{t=1}^{T}(f_t(\boldsymbol{w}_t) - f_t(\boldsymbol{w}^*)) \leq \sum_{t=1}^{T} \langle \boldsymbol{w}_t - \boldsymbol{w}^*, \boldsymbol{g}_t \rangle$$

$$\leq \frac{1}{1-\beta_1}\sum_{t=1}^{T}\left[\frac{1}{2\hat{\alpha}_t}(\|\sqrt{\boldsymbol{v}_t}(\boldsymbol{w}_t - \boldsymbol{w}^*)\|^2 - \|\sqrt{\boldsymbol{v}_t}(\boldsymbol{w}_{t+1} - \boldsymbol{w}^*)\|^2) + \frac{\beta_{1,t}}{2\hat{\alpha}_t}\|\boldsymbol{w}_t - \boldsymbol{w}^*\|^2\right.$$

$$\left. + \frac{\beta_{1,t}\hat{\alpha}_t}{2}\|\boldsymbol{m}_{t-1}\|^2 + \frac{\hat{\alpha}_t}{2}\|\frac{\boldsymbol{m}_t}{\sqrt{\boldsymbol{v}_t}}\|^2\right],$$

(16)

where the first inequality follows from the convexity of $f_t(\boldsymbol{w})$. For further bounding Equation (16), we need the following lemma.

**Lemma 4.** *For the parameter settings and conditions assumed in Theorem 2, we have*

$$\sum_{t=1}^{T} \hat{\alpha}_t \|\boldsymbol{m}_t\|^2 < \frac{2n\alpha G_\infty^2}{(1-\beta_1)^3}\sqrt{T}.$$

*Proof.* From Equation (4), we have

$$\hat{\alpha}_t \|\boldsymbol{m}_t\|^2 = \hat{\alpha}_t \| \sum_{i=1}^{t} (1 - \beta_{1,t-i+1}) \boldsymbol{g}_{t-i+1} \prod_{j=1}^{i-1} \beta_{1,t-j+1} \|^2 \leq \hat{\alpha}_t \| \sum_{i=1}^{t} \boldsymbol{g}_{t-i+1} \beta_1^{i-1} \|^2$$

$$= \hat{\alpha}_t \sum_{j=1}^{n} \left( \sum_{i=1}^{t} g_{t-i+1,j} \beta_1^{i-1} \right)^2 \leq \hat{\alpha}_t \sum_{j=1}^{n} \left( \sum_{i=1}^{t} g_{t-i+1,j}^2 \beta_1^{i-1} \right) \left( \sum_{i=1}^{t} \beta_1^{i-1} \right)$$

$$< \frac{\alpha}{\sqrt{t}} \frac{1}{1 - \beta_{1,t}} \frac{nG_\infty^2}{(1-\beta_1)^2} \leq \frac{n\alpha G_\infty^2}{(1-\beta_1)^3} \frac{1}{\sqrt{t}},$$

where the second inequality follows from Cauchy-Schwartz inequality. Therefore,

$$\sum_{t=1}^{T} \hat{\alpha}_t \|\boldsymbol{m}_t\|^2 < \frac{n\alpha G_\infty^2}{(1-\beta_1)^3} \sum_{t=1}^{T} \frac{1}{\sqrt{t}} < \frac{2n\alpha G_\infty^2 \sqrt{T}}{(1-\beta_1)^3},$$

where the last inequality follows from

$$\sum_{t=1}^{T} \frac{1}{\sqrt{t}} = 1 + \int_2^3 \frac{1}{\sqrt{2}} ds + \cdots + \int_{T-1}^{T} \frac{1}{\sqrt{T}} ds$$

$$< 1 + \int_2^3 \frac{1}{\sqrt{s-1}} ds + \cdots + \int_{T-1}^{T} \frac{1}{\sqrt{s-1}} ds$$

$$= 1 + \int_2^T \frac{1}{\sqrt{s-1}} ds = 2\sqrt{T-1} - 1 < 2\sqrt{T}.$$

This completes the proof. $\qquad \square$

Now we return to the proof of Theorem 2. Let $\hat{\alpha}_0 := \hat{\alpha}_1$. By Lemma 1, Lemma 2, Lemma 4, Equation (16) and the third inequality of Equation (6), we have

$$\sum_{t=1}^{T} (f_t(\boldsymbol{w}_t) - f_t(\boldsymbol{w}^*)) \leq \frac{1}{1-\beta_1} \left[ \frac{1}{2\hat{\alpha}_1} \|\sqrt{\boldsymbol{v}_1}(\boldsymbol{w}_1 - \boldsymbol{w}^*)\|^2 + \sum_{t=2}^{T} \left( \frac{1}{2\hat{\alpha}_t} \|\sqrt{\boldsymbol{v}_t}(\boldsymbol{w}_t - \boldsymbol{w}^*)\|^2 - \frac{1}{2\hat{\alpha}_{t-1}} \|\sqrt{\boldsymbol{v}_{t-1}}(\boldsymbol{w}_t - \boldsymbol{w}^*)\|^2 \right) \right.$$

$$+ \sum_{t=1}^{T} \frac{\beta_{1,t}}{2\hat{\alpha}_t} \|\boldsymbol{w}_t - \boldsymbol{w}^*\|^2 + \sum_{t=1}^{T} \left( \frac{\hat{\alpha}_{t-1}}{2} \|\boldsymbol{m}_{t-1}\|^2 + \frac{\hat{\alpha}_t}{2} \|\frac{\boldsymbol{m}_t}{\sqrt{\boldsymbol{v}_t}}\|^2 \right) \right]$$

$$\leq \frac{1}{1-\beta_1} \left[ \frac{D_\infty^2}{2\hat{\alpha}_1} \|\sqrt{\boldsymbol{v}_1}\|^2 + \sum_{t=2}^{T} D_\infty^2 \left( \frac{\|\sqrt{\boldsymbol{v}_t}\|^2}{2\hat{\alpha}_t} - \frac{\|\sqrt{\boldsymbol{v}_{t-1}}\|^2}{2\hat{\alpha}_{t-1}} \right) + \sum_{t=1}^{T} \frac{\beta_{1,t}}{2\hat{\alpha}_t} nD_\infty^2 \right.$$

$$+ \sum_{t=1}^{T} \hat{\alpha}_t \left( \frac{1}{2} + \frac{1}{2\delta\sqrt{1-\beta_2}} \right) \|\boldsymbol{m}_t\|^2 \right]$$

$$= \frac{1}{1-\beta_1} \left[ D_\infty^2 \frac{\|\sqrt{\boldsymbol{v}_T}\|^2}{2\alpha_T} + \sum_{t=1}^{T} \frac{\beta_{1,t}}{2\hat{\alpha}_t} nD_\infty^2 + \sum_{t=1}^{T} \hat{\alpha}_t \left( \frac{1}{2} + \frac{1}{2\delta\sqrt{1-\beta_2}} \right) \|\boldsymbol{m}_t\|^2 \right]$$

$$< \frac{1}{1-\beta_1} \left[ \frac{n(2G_\infty + \delta)D_\infty^2}{2\alpha\sqrt{1-\beta_2}(1-\beta_1)^2} \sqrt{T} + \sum_{t=1}^{T} \frac{\beta_{1,t}}{2\hat{\alpha}_t} nD_\infty^2 + \frac{n\alpha G_\infty^2}{(1-\beta_1)^3} \left( 1 + \frac{1}{\delta\sqrt{1-\beta_2}} \right) \sqrt{T} \right].$$

This completes the proof. $\qquad \square$

# E  Proof of Corollary 3

*Proof.* Since $\beta_{1,t} = \beta_1/t$, we have

$$\sum_{t=1}^{T} \frac{\beta_{1,t}}{2\hat{\alpha}_t} = \sum_{t=1}^{T} \frac{(1 - \beta_{1,t}^t)\sqrt{t}\beta_{1,t}}{2\alpha\sqrt{1-\beta_2^t}} < \sum_{t=1}^{T} \frac{\sqrt{t}\beta_{1,t}}{2\alpha\sqrt{1-\beta_2}} = \frac{\beta_1}{2\alpha\sqrt{1-\beta_2}} \sum_{t=1}^{T} \frac{1}{\sqrt{t}} < \frac{\beta_1}{\alpha\sqrt{1-\beta_2}} \sqrt{T}.$$

This completes the proof. $\qquad \square$

