# OpenReview forum: "AGD: an Auto-switchable Optimizer using Stepwise Gradient Difference for Preconditioning Matrix"
_NeurIPS.cc/2023/Conference — NeurIPS 2023 poster_

### Official Review · Reviewer_BE2A · 2023-07-05

**Soundness:** 3 good
**Presentation:** 3 good
**Contribution:** 2 fair
**Rating:** 7
**Confidence:** 2

**Summary:**

The paper proposes a new gradient based optimization called AGD. The idea is to utilize gradient difference between the current and previous steps in the preconditioning matrix which is related to Hessian. Also, an autoswitching function is proposed to switch between SGD and the adaptive optimizer. As mentioned in the paper, both ideas were previously brought up but the attempt to make an explicit optimizer is first done in the paper.

**Strengths:**

- Well written and motivated
- Conducting number of downstream experiments
- Presenting extra studies to examine the method

**Weaknesses:**

-The idea is not fully new
- The details of experiments could be more explained

**Questions:**

- Figure 3 suggests AGD is faster to converge, and Table 3 suggests AGD results in better performance. Could we make such a conclusion that it should be always used instead of SOTA optimization methods?

- What is the stop criteria of different optimization techniques in Table 3? I am wondering if the improvement obtained by AGD is accurately examined or for example all methods go for the same number of iterations and AGD converges faster? You argued this issue in section 4.3 for CV task.

- The performance of AGD in CV task is quite marginal. How about the convergence time? does AGD maybe converges faster?

- I would have also liked to see more about autoswitching and why and how it happens in a simple toy example.

**Limitations:**

-

---

> ### Author Rebuttal · Authors · 2023-08-08
>
> Thanks for your valuable review.
>
> **Q1: Figure 3 suggests AGD is faster to converge, and Table 3 suggests AGD results in better performance. Could we make such a conclusion that it should be always used instead of SOTA optimization methods?**
>
> AGD is at least the optimal method in the datasets and models we have tested. We believe it will also have promising performance in other scenarios.
>
> **Q2: What is the stop criteria of different optimization techniques in Table 3? I am wondering if the improvement obtained by AGD is accurately examined or for example all methods go for the same number of iterations and AGD converges faster? You argued this issue in section 4.3 for CV task.**
>
> In regards to the experiments detailed in Table 3, both the LSTM and transformer models were trained for a consistent number of epochs until convergence (200 epochs for LSTM and 55 epochs for transformers). With the exception of the optimizer hyperparameters mentioned in Appendix A.1, such as learning rate, epsilon, and delta, all other training settings remained identical across optimizers. These settings were inherited from previously published works to ensure a fair and credible comparison.
>
> **Q3: The performance of AGD in CV task is quite marginal. How about the convergence time? does AGD maybe converges faster?**
>
> The convergence speed of different optimizers can be found in Fig. 6 of Appendix A.2. For ResNet18 on ImageNet, AGD has a faster convergence speed compared to other optimizers. For ResNet20 and ResNet32 on Cifar10, after the learning rate decay, AGD demonstrates better convergence compared to other optimizers except for AdaHessian.
>
> **Q4: I would have also liked to see more about autoswitching and why and how it happens in a simple toy example.**
>
> Consider using AGD to optimize the function $f(x, y) = |x| + |y|$ from the starting point $(-4, 0.5)$, see Figure 4 in the [provided PDF](https://openreview.net/attachment?id=wLS9DFtY0I&name=pdf). Choose hyperparameters of $lr = 1.0$, $\delta = 1.0$, and $(\beta_1, \beta_2) = (0.0, 0.0)$. In the X-axis direction, the gradient difference is 0, so taking the maximum allows the parameters to be updated in an SGD-like manner. In the Y-axis direction, the gradient difference is 2, which is greater than 1, so the parameters are updated in an adaptive manner. Auto switch allows the model to use different update methods in different directions, and switches adaptively between SGD and adaptive methods as training progresses.
>
> We hope our responses have addressed your concerns.

---

> > ### Comment · Reviewer_BE2A · 2023-08-15
> > **after rebuttal**
> >
> > Thanks for the response!

---

### Official Review · Reviewer_ns5n · 2023-07-05

**Soundness:** 2 fair
**Presentation:** 3 good
**Contribution:** 2 fair
**Rating:** 3
**Confidence:** 4

**Summary:**

This paper introduces a novel optimizer, namely, Auto-switchable optimizer with Gradient Difference of adjacent steps. The authors propose a different way of approximating information from the Hessian for faster and better convergence in a way that a preconditioning matrix is computed using successive optimization steps in the parameter steps. The authors evaluate their optimization algorithm across different tasks and including some widely used architectures such as Transformers and RNNs. Moreover, the authors provide a theoretical analysis of the convergence of their algorithm and some bounds.


**Strengths:**

- The paper is well written in terms of giving the appropriate context of the most commonly used optimization algorithms, nice figures for toy experiments to understand how the algorithm works.
- The experimental framework idea is well designed in the sense that a good range of optimization tasks is considered to show the importance of the algorithm.
- The idea of using the successive steps seems novel to me to approximate Hessian information for faster convergence.


**Weaknesses:**

- In my humble opinion, the issue with the current idea lies in terms of the number/width of the optimization steps and how far the solution lies compared to the initialization point in the parameter space. The information that one can obtain by gradient differentials in the optimization space is only getting less and less useful while we increase the dimensionality of the optimization space. That means that for smaller optimization problems where the optimal solution would lie close to the initialization point, then I can really see the benefit of AGD compared to less Hessian well-informed optimizers like Adam. One can also see this emergent problem even from the theoretical analysis of the authors. For instance, in Theorem 2, the authors bound the approximation of the function using $D_\infty$ which effectively is the upper bound in terms of successive optimization steps in the parameter space and also an upper bound of each weight vector compared to its distance from the optimal solution. That means that for real-world multi-dimensional optimization problems (e.g. billions of parameters) those bounds and probably the algorithm itself could be called into question. Having said that, there are other works like NTK that have shown that neural networks stay pretty close in general near the initialization regime, so, I would not consider it appropriate to ask the authors to prove these things but they could put more effort into making their experiments more convincing to ameliorate this huge concern.
- Building upon my previous argument, my belief about the problem of scaling of the algorithm to bigger optimization problems is also displayed in the experiments where AGD scores significantly higher than all the other optimizers in smaller parameter spaces (e.g. 1-2 layers of LSTM) but becomes almost negligible with slightly bigger networks (e.g. compare Adam vs Proposed in Table 4, ResNEt 20, 32, Table 5 MLP, DCN, compare AdamW vs AGD in Table 1 Transformer). Having said that, the authors can include experiments with larger and state-of-the-art networks (e.g. ResNet 100+ layers on ImageNet, include some experiments with some networks with 500+ million parameters transformers) to prove me wrong and compare against state-of-the-art performance reported numbers with SGD and/or Adam.

I am willing to increase my score if most of the most important above concerns are addressed by the authors since I truly believe that the paper has a great potential to offer a more robust general optimization strategy to replace Adam.


**Questions:**

Would you consider including experiments with some harder tasks such as estimation tasks, e.g. audio source separation instead of detection and retrieval tasks?


**Limitations:**

The authors should include my first point under the weaknesses of the paper and include a small paragraph to address this potential limitation of their algorithm.

---

> ### Author Rebuttal · Authors · 2023-08-08
>
> Thanks for your constructive review.
>
> **Q1: In my humble opinion, the issue with the current idea lies in terms of the number/width of the optimization steps and how far the solution lies compared to the initialization point in the parameter space ... That means that for real-world multi-dimensional optimization problems (e.g. billions of parameters) those bounds and probably the algorithm itself could be called into question ... I would not consider it appropriate to ask the authors to prove these things but they could put more effort into making their experiments more convincing to ameliorate this huge concern.**
>
> To our knowledge, the issue exists for all optimizers. The generalization bound of Adam-like optimizers also includes $D_\infty$, as shown in Theorem 5 of [1]. The generalization bound of SGD includes $\\| x_0 - x^* \\|^2$, where $x_0$ is the starting point and $x^*$ is the optimal point, as shown in Theorem 2.1.14 of [2]. Therefore, the statement `the issue with the current idea lies in ... how far the solution lies compared to the initialization point in the parameter space` is inappropriate.
>
> In addition, we have experimentally demonstrated the effectiveness of AGD for models with a large number of parameters, as shown in the Figure 1 of the [PDF provided](https://openreview.net/attachment?id=wLS9DFtY0I&name=pdf) in the [Global Response](https://openreview.net/forum?id=A954O4tDmU&noteId=wLS9DFtY0I). We conducted optimizer experiments on the GPT2 (124 M) and GPT2-Large (774 M) models using the OpenWebText dataset. The results indicate that AGD outperformed the AdamW optimizer in terms of convergence. We hope our results address your concern.
>
> **Q2: My belief about the problem of scaling of the algorithm to bigger optimization problems is also displayed in the experiments where AGD scores significantly higher than all the other optimizers in smaller parameter spaces (e.g. 1-2 layers of LSTM) but becomes almost negligible with slightly bigger networks (e.g. compare Adam vs Proposed in Table 4, ResNEt 20, 32, Table 5 MLP, DCN, compare AdamW vs AGD in Table 1 Transformer).**
>
> Firstly, LSTMs tend to have far more parameters than ResNets, as convolutional kernels typically have fewer parameters. This difference can be observed in Table 2 of the paper.
>
> Additionally, the improvement in metrics can vary depending on the model’s capacity and the specific dataset being used. For instance, enhancing the AUC of ReSys tasks is known to be challenging, and certain networks are often over-parameterized for smaller datasets (e.g., ResNets on Cifar10). Consequently, achieving improvement from the optimizer’s perspective can be challenging but valuable as well.
>
> Lastly, we have conducted extensive experiments of AGD on GPTs since the rise of LLMs, and the results have been highly promising, as indicated in the response to Q1. The pretraining stage of LLMs can typically be time-consuming, often spanning several months. However, we are confident that AGD has the potential to considerably expedite the training of LLMs.
>
> **Q3: Having said that, the authors can include experiments with larger and state-of-the-art networks (e.g. ResNet 100+ layers on ImageNet, include some experiments with some networks with 500+ million parameters transformers) to prove me wrong and compare against state-of-the-art performance reported numbers with SGD and/or Adam.**
>
> Please refer to the [Global Response 1](https://openreview.net/forum?id=A954O4tDmU&noteId=wLS9DFtY0I). Also see Q1.
>
> **Q4: Would you consider including experiments with some harder tasks such as estimation tasks, e.g. audio source separation instead of detection and retrieval tasks?**
>
> Following [3, 4], we test optimizers against common CV and NLP tasks. We will leave the harder tasks you mentioned for future work.
>
> We hope our responses have addressed your concerns.
>
> ---
> References:
>
> [1] Reddi, S. J. et al., On the Convergence of Adam and Beyond. ICLR 2019.
>
> [2] Yurii E. Nesterov. Introductory Lectures on Convex Optimization - A Basic Course, volume 87 of Applied Optimization. Springer, 2004.
>
> [3] Zhuang, J. et al., AdaBelief Optimizer: Adapting Stepsizes by the Belief in Observed Gradients. NeurIPS 2020.
>
> [4] Yao, Z. et al., ADAHESSIAN: An Adaptive Second Order Optimizer for Machine Learning. AAAI 2021.

---

> > ### Comment · Reviewer_ns5n · 2023-08-15
> > **Response to the authors.**
> >
> > Thanks for your response and for trying to address my concerns. Here is my response:
> >
> > > To our knowledge, the issue exists for all optimizers. The generalization bound of Adam-like optimizers also includes , as shown in Theorem 5 of [1]. The generalization bound of SGD includes , where is the starting point and is the optimal point, as shown in Theorem 2.1.14 of [2]. Therefore, the statement the issue with the current idea lies in ... how far the solution lies compared to the initialization point in the parameter space is inappropriate.
> >
> > The statement is perfectly valid since the truth is that Adam and other optimizers are prefered over other optimizers only for their experimental results and not for their theoretical properties. Although Adam like optimizers might have the same issue in proving useful bounds, that fact alone does not mean that all other theoretical bounds for Adam-like optimizers should also prove bounds which are not informative. All in all though, the authors' answer is not correlated with my criticism which was: _"In my humble opinion, the issue with the current idea lies in terms of the number/width of the optimization steps and how far the solution lies compared to the initialization point in the parameter space. The information that one can obtain by gradient differentials in the optimization space is only getting less and less useful while we increase the dimensionality of the optimization space. That means that for smaller optimization problems where the optimal solution would lie close to the initialization point, then I can really see the benefit of AGD compared to less Hessian well-informed optimizers like Adam."_. In my initial review I also said: _"I would not consider it appropriate to ask the authors to prove these things but they could put more effort into making their experiments more convincing to ameliorate this huge concern."_, thus, I think the authors have misinterpreted my critisism and considered that I am attacking their proof whereas I was simply talking about the fact that their algorithm might provide boosts over vanilla Adam for smaller optimization spaces. Having said that, I acknoledge and welcome that the authors have tried to run the appropriate large scale experiments to show that my concern is not valid and empriically show that their algorithm outperforms AdamW.
> >
> > > In addition, we have experimentally demonstrated the effectiveness of AGD for models with a large number of parameters, as shown in the Figure 1 of the PDF provided in the Global Response. We conducted optimizer experiments on the GPT2 (124 M) and GPT2-Large (774 M) models using the OpenWebText dataset. The results indicate that AGD outperformed the AdamW optimizer in terms of convergence. We hope our results address your concern.
> >
> > Thanks for running this experiment. However, the validation losses reported on https://github.com/karpathy/nanoGPT when training these larger models on OpenWebText are 3.12 and 2.67 for GPT-2 and GPT-2-LARGE, respectively. In the authors' rebuttal, the reported baseline numbers are approximately 3.08 and 2.72 for GPT-2 and GPT-2-LARGE, correspondingly. That fact makes me skeptical about the validity of these experiments and how can a discrepancy like this be caused (I would expect both losses to be higher than the reported ones in https://github.com/karpathy/nanoGPT in the case of less training time or at least have a similar pattern). It could be the case that the weight decay could play a role here and Adam would also behave very similarly to AGD.
> >
> > >   Firstly, LSTMs tend to have far more parameters than ResNets, as convolutional kernels typically have fewer parameters. This difference can be observed in Table 2 of the paper.
> >
> > Firstly, the above statement is absolutely incorrect, the number of parameters in convolutional and recurrent neural networks depends on the task and the context that one wants to model. It might be true that for some easy problems like image classification for 28x28 pixels, one only needs a few small convolutional kernels in order to obtain the appropriate receptive field. On the contrary, if you had an audio waveform sampled at 16kHz and you wanted to model time-dependencies over 5 second clips, one would need a much larger amount of trainable parameters for a CNN model compared to an LSTM that can implicitly carry the state for each time-step. In Table 2, the authors compare LSTM models for NLP tasks and compare them with CNNs for computer vision tasks with input-sizes of at most (224x224), thus, no generalization can be made from this Table.
> >
> > All in all, the authors' experiments with GPT make me skeptical and my main criticism (AGD behaves similarily to Adam except of smaller optimization spaces) remains. Thus, I cannot increase my score until I see a clear performance improvement of AGD vs Adam for a large-scale model and task.

---

> > > ### Author Response · Authors · 2023-08-16
> > > **Clarification to the confusion on GPT-2**
> > >
> > > Thank you for your response. I believe the main concern revolves around the performance of GPT-2. The README of nanoGPT states the following:
> > >
> > > >  However, we have to note that GPT-2 was trained on (closed, never released) WebText, while OpenWebText is just a best-effort open reproduction of this dataset. This means there is a dataset domain gap. Indeed, taking the GPT-2 (124M) checkpoint and finetuning on OWT directly for a while reaches loss down to **~2.85**. This then becomes the more appropriate baseline w.r.t. reproduction.
> > >
> > > The numbers you mentioned are evaluations conducted on OpenWebText using OpenAI’s checkpoint, which is trained on the closed WebText dataset. It is expected that the val loss would be higher due to domain shift. The repo author also mentions the nanoGPT's actuall loss (**~2.85**) in [this section](https://github.com/karpathy/nanoGPT/tree/master#reproducing-gpt-2). Our result for GPT-2 with 50000 steps, approximately 3.08, is larger than 2.85.
> > >
> > > For testing, we keep the weight decay and other training settings the same, except for the learning rate. We choose lr as 6e-5/1e-4 for GPT-2/GPT2-Large, and delta as 1e-14 based on our experience with transformers small from paper. It is worth mentioning that though AGD is not well tuned, the results are promising.
> > >
> > > I hope this would clarify your confusion and address your concern about AGD's performance on large scale models.

---

> > > > ### Comment · Reviewer_ns5n · 2023-08-16
> > > > **Response 2**
> > > >
> > > > Thanks for the clarification. It seems that indeed the numbers reported in https://github.com/karpathy/nanoGPT/tree/master#reproducing-gpt-2 refer to the pre-trained OpenAI checkpoints. Given that the latter is true, that means that in your experiments you managed to obtain a value of ~2.67 for the train loss and ~2.72 for the validation loss using AdamW and the same parameters except of the learning rate which is almost as good as the ones reported on the github page. Thus, this means that this is not training from scratch and is simply a finetuning of a pre-trained checkpoint if I am not confused again. If the latter is true, that means that this is not the approrpiate experiment to show that AGD outperforms Adam in a large-scale optimization setup and task especially when you 1) compare with a method that has weight decay and not Adam and compare how fast the methods converge 2) you change the default learning rate to some other empirical value 3) finetuning is a way easier problem than starting from scratch and does not bear the same empirical validity as a full-scale experiment.
> > > >
> > > > Again, the experiments with GPT make me skeptical and my main criticism (AGD behaves similarily to Adam except of smaller optimization spaces) remains. Thus, I cannot increase my score until I see a clear performance improvement of AGD vs Adam for a large-scale model and task.

---

> > > > > ### Author Response · Authors · 2023-08-16
> > > > > **Clarification on the difference between OpenAI GPT-2/nanoGPT**
> > > > >
> > > > > Thanks for the response.
> > > > >
> > > > > Our experiments are **training from scratch** following the [reproducing GPT-2](https://github.com/karpathy/nanoGPT#reproducing-gpt-2) part. The confusing part is here:
> > > > >
> > > > > >  This will run for about 4 days using PyTorch Distributed Data Parallel (DDP) and go down to loss of **~2.85**. Now, a GPT-2 model just evaluated on OWT gets a val loss of about 3.11, but if you finetune it it will come down to **~2.85** territory (due to an apparent domain gap), making the two models ~match.
> > > > >
> > > > > Simply put, a well-trained GPT-2 (124M) with nanoGPT using AdamW has a loss of ~2.85, which is exactly what an OpenAI checkpoint finetuned with OpenWebText can achieve. Following this observation, the repo author proposes [baselines](https://github.com/karpathy/nanoGPT#baselines) of **direct loss calculations** of OpenAI checkpoint on OpenWebText train/val in the following section, which should be **larger** than loss of OpenAI checkpoint finetuned on OWT, equally the final loss of training from scratch using nanoGPT.
> > > > >
> > > > > For the GPT-2-Large model, the val set loss of OpenAI checkpoint is 2.67, which means if we finetune this checkpoint or train from scratch using nanoGPT, we can achieve **a lower loss**. As for the number you bring up, `~2.72 for the validation loss using AdamW`, is fair because it is in the middle of convergence.
> > > > >
> > > > > The repo also contains a train/val loss curve at top for GPT-2 (124M), which provides a better view of training dynamics.
> > > > >
> > > > > I find it a little confusing at first, hope my explanation can help clarify it for you.

---

> > > > > > ### Comment · Reviewer_ns5n · 2023-08-16
> > > > > > **Response 3**
> > > > > >
> > > > > > Thanks for the clarification. It could help if you had a detailed explanation of the large-scale ecale experiment you ran and compare it with some reference from the literature (certainly a github page is not ideal but in these days maybe it is better than many papers out there). Anyways, considering that whatever you said is correct and you are training from scratch, the comparison you are making is still not valid IMHO for 2 out of the 3 reasons which I explained in my previous message:
> > > > > >
> > > > > > > If the latter is true, that means that this is not the approrpiate experiment to show that AGD outperforms Adam in a large-scale optimization setup and task especially when you 1) compare with a method that has weight decay and not Adam and compare how fast the methods converge 2) you change the default learning rate to some other empirical value ~3) finetuning is a way easier problem than starting from scratch and does not bear the same empirical validity as a full-scale experiment.~

---

> > > > > > > ### Author Response · Authors · 2023-08-16
> > > > > > > **More details on GPT-2**
> > > > > > >
> > > > > > > Thanks for response.
> > > > > > >
> > > > > > > Basically, nanoGPT is an open source replica of GPT-2 [1], for language modeling. [2] is benchmarking with different implementation and dataset, but we choose nanoGPT due to its simplicity and reproductivity.
> > > > > > >
> > > > > > > The shared config for AdamW/AGD is [train_gpt2.py](https://github.com/karpathy/nanoGPT/blob/master/config/train_gpt2.py) from the repo. Note `weight_decay=1e-1` is kept identical for AdamW/AGD as I mentioned before, because the decoupled weight decay technique is the de-facto standard especially when training transformers. So in paper and here, we do not explicitly refer AGD with weight decay as AGDW, as others do not either. Hope this resolves your concern 1.
> > > > > > >
> > > > > > > For your concern 2, different optimizers should have different optimal learning rates, especially when they are distinct by design. As for AdamW and AGD, they have apparent different precondition matrices, which leads to disagreement on lr.  The lr for AdamW is 6e-4, and AGD outperforms AdamW with a smaller lr.
> > > > > > >
> > > > > > > ---
> > > > > > > References:
> > > > > > >
> > > > > > > [1] Radford, A., Wu, J., Child, R., Luan, D., Amodei, D., & Sutskever, I. (2019). Language models are unsupervised multitask learners.
> > > > > > >
> > > > > > > [2] Luo, Y., Ren, X., Zheng, Z., Jiang, Z., Jiang, X., & You, Y. (2023). CAME: Confidence-guided Adaptive Memory Efficient Optimization.

---

### Official Review · Reviewer_1Tit · 2023-07-11

**Soundness:** 2 fair
**Presentation:** 2 fair
**Contribution:** 3 good
**Rating:** 5
**Confidence:** 3

**Summary:**

The paper proposes a new optimizer based on finite difference approximation to obtain the inner product between the Hessian row vector and the parameter vector difference from gradients of succeeding steps and an auto-switching function to switch between SGD and the adaptive optimizer. It uses an exponential moving average of gradient instead of gradient for lower variance. Experimental results show that the proposed method performs better than existing methods in most cases.

**Strengths:**

The proposed algorithm can efficiently acquire the information of the Hessian.

Experimental results show performance improvements.

**Weaknesses:**

The explanation of related works is minimum. It needs to be more concretely clarified what problem of existing second-order methods it addresses.

The experiments in section 4 only report model performance and do not report computational costs.

I need help finding Figures 5 and 6 in the paper. Are they missing?

**Questions:**

What is the reason for the performance improvement? Is it due to its ability to find better local optima or the faster convergence in a limited computation budget?

**Limitations:**

There is no numerical explanation for the computational cost.

---

> ### Author Rebuttal · Authors · 2023-08-08
>
> Thanks for your thoughtful review.
>
> **Q1: The explanation of related works is minimum. It needs to be more concretely clarified what problem of existing second-order methods it addresses.**
>
> Second-order methods entail the computation or approximation of the Hessian, which can be computationally demanding. Despite efforts to reduce overhead, such as the Hessian diagonal approximation employed by AdaHessian, these methods are still less efficient compared to first-order methods. In response to Q2, we will present experimental results regarding the computational costs involved.
>
> The major problem of the existing second-order methods, like AdaHessian, is that they require extra memory footprints and heavy computation compared to first-order methods, which makes they unusable in practice. Please refer to [Global Response 2](https://openreview.net/forum?id=A954O4tDmU&noteId=wLS9DFtY0I), where we compare the resource usage experimentally.
>
> **Q2: The experiments in section 4 only report model performance and do not report computational costs.**
>
> In theory, AGD requires approximately the same amount of memory as AdamW, as they both store two optimizer states, with slightly higher computational requirements. In order to demonstrate this, we provide experimental results in [Global Response 2](https://openreview.net/forum?id=A954O4tDmU&noteId=wLS9DFtY0I).
>
> **Q3: I need help finding Figures 5 and 6 in the paper. Are they missing?**
>
> Thanks for pointing out this issue. Figures 5 and 6 are in Appendix A.2 due to space limitations. We will provide further clarification in the next revised version, as well as fix a typo in Figure 5.
>
> **Q4: What is the reason for the performance improvement? Is it due to its ability to find better local optima or the faster convergence in a limited computation budget?**
>
> Our findings suggest that the observed improvement in performance can be attributed to both fast convergence and good generalization. The fast convergence is primarily thanks to the gradient difference between the previous and current steps, which effectively incorporates Hessian information into the preconditioning matrix. Furthermore, the auto-switch mechanism we implemented can prevent abnormal updates and and, compared to the traditional method of adding a small amount, can avoid introducing additional bias during updates, thereby leading to more stable training and improved generalization.
>
> We hope our responses have addressed your concerns.

---

### Official Review · Reviewer_Ag5C · 2023-07-13

**Soundness:** 3 good
**Presentation:** 3 good
**Contribution:** 3 good
**Rating:** 7
**Confidence:** 3

**Summary:**

The paper proposes a new optimizer called AGD that integrates the information of the Hessian into the preconditioning matrix and switches dynamically between SGD and the adaptive optimizer. The authors establish theoretically proven convergence guarantees in both non-convex and convex stochastic settings. The authors validate AGD on a total of six public datasets: two from NLP (IWSLT14 and PTB), two from CV (Cifar10 and ImageNet), and the rest from RecSys (Criteo and Avazu). The experimental results reveal that AGD can outperform or be on a par with the SOTA optimizers. Overall, the paper makes a significant contribution to the field of deep learning optimization. AGD is a promising new optimizer that has the potential to improve the performance of deep learning models on a variety of tasks.

**Strengths:**

• AGD can achieve a faster convergence rate over SGD

• AGD can automatically switch between stochastic and adaptive optimization, depending on the progress of the optimization. This allows AGD to achieve the best of both worlds, i.e., the fast convergence of adaptive optimizers and the robustness of SGD.

• AGD has a few flexible hyperparameters, making tuning for different tasks and datasets easy.

• AGD was evaluated on various datasets across different domains, and it outperformed other optimizers in most cases.


**Weaknesses:**

• The convergence rate of AGD depends on the hyperparameters δ and β1. If these hyperparameters are not chosen carefully, AGD may not converge to the optimal solution. Looking forward to seeing more comprehensive robustness evaluation using different network structures and larger datasets.

• AGD requires more computation than SGD. How is its efficiency for large-scale problems, e.g., large language models? I think performance on large-scale generative model optimization is a very interesting direction. Does the author have plans to try on such a task?


**Questions:**

•	Please see weaknesses.

**Limitations:**

There is no limitation discussed in the paper.

---

> ### Author Rebuttal · Authors · 2023-08-08
>
> Thanks for your insightful review.
>
> **Q1: The convergence rate of AGD depends on the hyperparameters δ and β1. If these hyperparameters are not chosen carefully, AGD may not converge to the optimal solution. Looking forward to seeing more comprehensive robustness evaluation using different network structures and larger datasets.**
>
> We also conducted robustness tests on $\delta$ for the ResNet18 model using the ImageNet dataset, and the results are shown in Figure 2 in the [provided PDF](https://openreview.net/attachment?id=wLS9DFtY0I&name=pdf). As can be seen, the model performance did not show significant changes within a large range of $\delta$ values. As a common parameter for adaptive optimizers, $\beta_1$ was not adjusted in this work in order to maintain consistency with other optimizers.
>
> **Q2: AGD requires more computation than SGD. How is its efficiency for large-scale problems, e.g., large language models? I think performance on large-scale generative model optimization is a very interesting direction. Does the author have plans to try on such a task?**
>
> Please refer to [Global Response 1](https://openreview.net/forum?id=A954O4tDmU&noteId=wLS9DFtY0I).
>
> We hope our responses have addressed your concerns.

---

### Official Review · Reviewer_xqym · 2023-07-26

**Soundness:** 3 good
**Presentation:** 3 good
**Contribution:** 3 good
**Rating:** 7
**Confidence:** 4

**Summary:**

This paper proposed AGD, a novel optimiser that can dynamically switch between an adaptive optimiser and SGD. To achieve this, an auto-switching function is introduced to change the diagonal elements of the preconditioning matrix based on the gradients of adjacent steps. Both theoretical analysis on the convergence of the optimiser, and experimental results on six public CV, NLP and RecSys setups are provided, demonstrating AGD is a promising method both in theory and practice.


**Strengths:**

It is often noticed in practice that SGD with proper hyperparameters (learning rate, momentum, and weight decay) can outperform the adaptive optimisers with better local optimum found, while the adaptive optimisers require less effort in hyperparameter tuning and sometimes faster convergence. The auto-switching strategy may help combine the advantages of them, and experimental results and analysis demonstrate the soundness and robustness of the chosen auto-switching function.


**Weaknesses:**

1. Optimisers can perform differently on large-scale and small-scale experiments. Although good PTB results are given in the paper, a larger-scale language modelling experiment may still be useful.
2. Some hyperparameter settings, such as those for PTB, are missing in the paper and the supplementary materials. The effect of regularisation methods, in particular L2 and dropout, are not given. This often makes a key difference to the generalisation ability of the test set and the performance of SGD and could be at least considered in the numerical analysis session.
3. The proof of the faster convergence speed of AGD using the case studies in Sec. 3.2 may have resulted in some questions, as the comparisons of optimisers with different learning rates and other hyperparameters may not be fair enough.
4. Extra memory cost, computational cost and possible limitations of the method are not discussed.


**Questions:**

What's the trajectory of Adam at the minimum point B in Figure 1?

**Limitations:**

Discussions of regularisation methods (at least with SGD) and large-scale experiments with complex model structures (e.g. a Transformer with both encoder and decoder) are missing.
The analysis of the convergence speed in the numerical analysis session may not be completely fair.
Would be useful to include the analysis of computation and storage costs.

---

> ### Author Rebuttal · Authors · 2023-08-08
>
> Thanks for your constructive review.
>
> **Q1: Optimisers can perform differently on large-scale and small-scale experiments. Although good PTB results are given in the paper, a larger-scale language modelling experiment may still be useful.**
>
> Please refer to [Global Response 1](https://openreview.net/forum?id=A954O4tDmU&noteId=wLS9DFtY0I).
>
> **Q2: Some hyperparameter settings, such as those for PTB, are missing in the paper and the supplementary materials. The effect of regularisation methods, in particular L2 and dropout, are not given. This often makes a key difference to the generalisation ability of the test set and the performance of SGD and could be at least considered in the numerical analysis session.**
>
> The training settings are consistent across all optimizers, including regularization methods. The specific training details for PTB are provided in the file nlp-task/lstm/run_all_layer{1,2,3}.py, where the general dropout rate is set to 0.4 and the weight decay is set to 1.2e-6 (as mentioned in Appendix A.1). We will provide more details in the revised version.
>
> **Q3: The proof of the faster convergence speed of AGD using the case studies in Sec. 3.2 may have resulted in some questions, as the comparisons of optimisers with different learning rates and other hyperparameters may not be fair enough.**
>
> As mentioned in Sec. 3.2, all hyperparameters are kept identical, including the learning rate of 1e-3 and betas of (0.9, 0.999). One exception is SGDM, which does not converge under such a large learning rate.
>
> **Q4: Extra memory cost, computational cost and possible limitations of the method are not discussed.**
>
> AGD and AdamW both store two optimizer states, so their memory footprints should be similar. AGD incurs a slight additional computational cost, which we validate on the transformer model and find to be negligible, especially when compared to optimizers like AdaHessian. The results are shown in [Global Response 2](https://openreview.net/forum?id=A954O4tDmU&noteId=wLS9DFtY0I).
>
> **Q5: What's the trajectory of Adam at the minimum point B in Figure 1?**
>
> The whole trajectory until convergence is shown in Figure 3 in the [provided PDF](https://openreview.net/attachment?id=wLS9DFtY0I&name=pdf).
>
> We hope our responses have addressed your concerns.

---

### Author Rebuttal · Authors · 2023-08-08

#### Global Response

1. **AGD performance on large models**

    We conduct optimizer experiments on the GPT2 (124M) and GPT2-Large (774M) models using the OpenWebText dataset. Our code is based on https://github.com/karpathy/nanoGPT. We compare AGD with the default AdamW optimizer. The results are shown in Figure 1 of the PDF provided. As can be seen, AGD outperforms AdamW even after both were run for 50,000 steps. We hope our results effectively address the concerns of the reviewers.

2. **AGD's computational cost**

    We train a transformer small model for IWSLT14 on a single Nvidia P100. AGD is comparable to the most commonly used optimizer AdamW, while significantly better than AdaHessian in terms of memory footprint and training speed. We will add these results to the next revision.

    | Optimier | Memory/Mb | Time per Epoch/s | Relative time to AdamW |
    | - | - | - | - |
    | SGDM | 5119 | 230 | 0.96× |
    | AdamW | 5413 | 260 | 1.00× |
    | AdaHessian | 8943 | 750 | 2.88× |
    | AGD | 5409 | 278 | 1.07× |


In addition, we have included some images in attachment PDF to address specific questions raised by the reviewers.

---

### Comment · Area_Chair_Azr5 · 2023-08-15

Dear reviewers,

The authors have uploaded their rebuttal.  Please take time to go over it.  If you have any further questions or concerns regarding the authors' rebuttal, please start a discussion.   If you are willing to adjust your scores after reading the rebuttal, please do.

Thanks,

AC

---

### Decision · Program_Chairs · 2023-09-21

**Decision:**

Accept (poster)

**Comment:**

This paper proposes an optimizer that can switch dynamically between SGD and being adaptive.  The authors design a diagonal preconditioning matrix where the diagonal elements are approximating the inner product between the Hessian row vector and the parameter vector difference of adjacent steps.  On top of that , depending on a threshold hyperparameter, the preconditioning matrix can switch between SGD and an adaptive one.   The proposed optimizer is evaluated on three tasks on a variety of datasets and models.   Superior convergence and generalization performance is reported and computation/memory cost is analyzed.  The authors also give a proof for its convergence and its rate.   The idea of auto-switching is interesting and extensive experiments are carried out to show its effectiveness in the original submission and rebuttal.  That being said, there are also issues with the papers. For instance, the authors assume bounded gradients in Theorem 1 and bounded set of W in Theorem 2, which are strong assumptions in optimization literature.  Furthermore, it is helpful to move some of the convergence curves to the main body rather than put all of them in supplementary as they are helpful to show the convergence behavior of an optimizer.   Overall,  given its strong empirical performance, this paper can be accepted.